# How good is Good-Turing for Markov samples?

**Prafulla Chandra**                                    *ee16d402@ee.iitm.ac.in*
*Department of Electrical Engineering, IIT Madras*
*Chennai, India*

**Andrew Thangaraj**                                    *andrew@ee.iitm.ac.in*
*Department of Electrical Engineering, IIT Madras*
*Chennai, India*

**Nived Rajaraman**                                    *nived.rajaraman@gmail.com*
*Department of Electrical Engineering and Computer Science*
*University of California*
*Berkeley, CA 94720-1776, USA*

**Reviewed on OpenReview:** *https://openreview.net/forum?id=KokkP2nQ24*

## Abstract

The Good-Turing (GT) estimator for the missing mass (i.e., total probability of missing symbols) in $n$ samples is the number of symbols that appeared exactly once divided by $n$. For i.i.d samples, the bias and squared-error risk of the GT estimator can be shown to fall as $1/n$ by bounding the expected error uniformly over all symbols. In this work, we study convergence of the GT estimator for missing stationary mass (i.e., total stationary probability of missing symbols) of Markov samples on an alphabet $\mathcal{X}$ with stationary distribution $[\pi_x : x \in \mathcal{X}]$ and transition probability matrix (t.p.m) $P$. This is an important and interesting problem because GT is widely used in applications with temporal dependencies such as language models assigning probabilities to word sequences, which are modelled as Markov. We show that convergence of GT depends on convergence of $(P^{\sim x})^n$, where $P^{\sim x}$ is $P$ with the $x$-th column zeroed out. This, in turn, depends on the Perron eigenvalue $\lambda^{\sim x}$ of $P^{\sim x}$ and its relationship with $\pi_x$ uniformly over $x$. For randomly generated t.p.ms and t.p.ms derived from New York Times and Charles Dickens corpora, we numerically exhibit such uniform-over-$x$ relationships between $\lambda^{\sim x}$ and $\pi_x$. This supports the observed success of GT in language models and practical text data scenarios. For Markov chains with rank-2, diagonalizable t.p.ms having spectral gap $\beta$, we show minimax rate upper and lower bounds of $1/(n\beta^5)$ and $1/(n\beta)$, respectively, for the estimation of stationary missing mass. This theoretical result extends the $1/n$ minimax rate for i.i.d or rank-1 t.p.ms to rank-2 Markov, and is a first such minimax rate result for missing mass of Markov samples. We also show, through experiments, that the MSE of GT decays at a slower rate as the rank of the t.p.m increases.

## 1 Introduction

When observing a sequence of symbols from an unknown alphabet and distribution, we are often interested in the probability that the next sampled symbol is going to be new, i.e. a symbol that has not been seen so far. This probability is the sum of the probabilities of all the symbols missing in the sequence of samples observed so far, and is called the *missing mass*. Good and Turing (Good, 1953) had originally studied this problem in the context of solving the enigma code. The popular Good-Turing (GT) estimator estimates missing mass as the ratio of the number of symbols seen exactly once in the samples to the sample size. Today, estimation of missing mass finds applications in language modelling (W.Church & A.Gale, 1991; Gale & Sampson, 1995; Chen & Goodman, 1996), ecology (Chao & Lee, 1992; Shen et al., 2003) and in entropy estimation (Vu et al., 2007), and it has been studied in the i.i.d samples setting by multiple authors (McAllester & Schapire, 2000;

Berend & Kontorovich, 2013; Chandra et al., 2019; Ohannessian & Dahleh, 2012; Mossel & Ohannessian, 2019; Orlitsky & Suresh, 2015; Rajaraman et al., 2017; Acharya et al., 2018; Cohen et al., 2020; Painsky, 2022). The mean-squared error for estimating missing mass using the GT estimator in the i.i.d setting (Rajaraman et al., 2017) falls as $(\text{sample-size})^{-1}$ with no further assumptions on the alphabet size or restrictions to the distribution. So, whenever the missing mass is expected to be non-vanishing, it can be reliably estimated in the i.i.d case.

While missing mass and the GT estimator are well-studied in the i.i.d samples regime, their definitions and properties in cases where the samples have memory have not been extensively considered in the literature. Many applications like natural language text processing involve data with temporal dependencies, which are often modelled as Markov chains (Chen & Goodman, 1996). In this work, we study missing mass and its estimation through GT estimator in cases when the samples form a Markov chain. Estimation from Markov samples has been considered in (Wolfer & Kontorovich, 2019; Hao et al., 2018; Han et al., 2018; Hsu et al., 2019; Han et al., 2021) and forays towards missing mass estimation from Markov chains were made in (Skorski, 2020; Chandra et al., 2020; Chandra et al., 2022). Though GT missing mass estimates are used as part of distribution estimation in cases where the samples have memory, theoretical properties of the estimation of missing mass in such scenarios is, to the best of our knowledge, considered for the first time here.

## 2 Missing stationary mass of a Markov chain

A sequence $X^n = (X_1, X_2, \ldots, X_n)$, $X_i \in \mathcal{X}$, is said to be a Markov chain if

$$\Pr(X_i{=}x_i | X_{i-1}{=}x_{i-1}, \ldots, X_1{=}x_1) = \Pr(X_2{=}x_i | X_1{=}x_{i-1})$$

for $i = 2, \ldots, n$ and all $x_i \in \mathcal{X}$. The $X_i$'s are called states and $\mathcal{X}$ is called the state space. $K \triangleq |\mathcal{X}|$ is the size of the alphabet $\mathcal{X}$. The transition probability matrix (t.p.m) of the Markov chain, denoted $P$, is the $K \times K$ matrix with $(i,j)$-th element $P_{ij} \triangleq \Pr(X_2 = j | X_1 = i)$. A distribution $\boldsymbol{\pi} = [\pi_1, \ldots, \pi_K]$ on $\mathcal{X}$ is said to be a stationary or invariant distribution of the Markov chain if $\boldsymbol{\pi}P = \boldsymbol{\pi}$ (Gallager, 1996). A Markov chain $X^n$ is said to be stationary if $X_1 \sim \boldsymbol{\pi}$, which implies that $X_i \sim \boldsymbol{\pi}$ for all $i$. We denote by $X^n \sim \text{Markov}(P, \boldsymbol{\pi})$, a stationary Markov chain with t.p.m. $P$ and state distribution $\boldsymbol{\pi}$.

Let $I(\cdot)$ and $E[\cdot]$ denote the indicator random variable and expectation, respectively, and let $[K]$ denote the set $\{1, 2, \ldots, K\}$. For $x \in \mathcal{X}$, $N_x(X^n) \triangleq \sum_{i=1}^{n} I(X_i = x)$ is the number of occurrences of $x$ in $X^n$, also called the frequency of $x$. For $l = 0, 1, 2, \ldots$, $\phi_l(X^n) \triangleq \sum_{x \in \mathcal{X}} I(N_x(X^n) = l)$ is the number of letters that have occurred $l$ times in $X^n$.

The *missing stationary mass* of a Markov chain $X^n \sim \text{Markov}(P, \boldsymbol{\pi})$, which is the missing mass of $\boldsymbol{\pi}$ in $X^n$, is defined as

$$M_0(\boldsymbol{\pi}, X^n) \triangleq \sum_{x \in \mathcal{X}} \pi_x \, I(N_x(X^n) = 0). \tag{1}$$

Estimation of the missing mass $M_0(\boldsymbol{\pi}, X^n)$ when $X^n \sim \text{Markov}(P, \boldsymbol{\pi})$ and the quantities $\mathcal{X}$, $K$, $\boldsymbol{\pi}$ and $P$ are unknown is important for many applications and in theory (see the role of missing mass in excess risk of competitive distribution estimation in (Orlitsky & Suresh, 2015)). Note that $M_0(\boldsymbol{\pi}, X^n)$ is a random variable that is a function of both the samples $X^n$ and the distribution $\boldsymbol{\pi}$. This makes estimation of missing mass and its analysis non-trivial even in the classical i.i.d regime where $X^n \sim$ i.i.d $\boldsymbol{\pi}$ or a Markov chain with $P = \mathbf{1}\boldsymbol{\pi}$. For $X^n \sim \text{Markov}(P, \boldsymbol{\pi})$ with a general $P$, the samples are not drawn exactly as per $\boldsymbol{\pi}$, which is the weight for measuring missing mass. This makes the study of missing stationary mass of a Markov chain more challenging, when compared to an i.i.d sequence.

The Good-Turing (GT) estimator (Good, 1953) for the missing mass $M_0(\boldsymbol{\pi}, X^n)$ is defined as

$$\widehat{M_0^{\text{GT}}}(X^n) \triangleq \frac{\phi_1(X^n)}{n}, \tag{2}$$

which is the fraction of symbols that have appeared exactly once in the $n$ samples.

The minimax squared-error risk[1] of estimating missing mass over a class of distributions $\mathcal{P}$, denoted $R_n^*(\mathcal{P})$, is defined as

$$R_n^*(\mathcal{P}) = \min_{\text{Estimator } \widehat{M_0}} \max_{(P, \boldsymbol{\pi}) \in \mathcal{P}} E_{X^n \sim \text{Markov}(P, \boldsymbol{\pi})}[(\widehat{M_0}(X^n) - M_0(\boldsymbol{\pi}, X^n))^2]. \tag{3}$$

The GT estimator has a worst-case squared error risk of $O(1/n)$ in the i.i.d regime and is known to be minimax rate-optimal (Rajaraman et al., 2017; Acharya et al., 2018). A common approach to use an estimator that works well on i.i.d sequences in the (non-i.i.d) Markov setting is to sub-sample the Markov chain at intervals of mixing time and apply the estimator on the resultant, nearly i.i.d, sub-sampled sequence. But such an estimate using the GT estimator will have a non-vanishing bias for the missing stationary mass of a Markov chain, since the missing mass of the subsampled sequence would be different and possibly greater than the missing mass of the whole Markov chain. Moreover, through examples, we demonstrate how the GT estimator fails to converge to the missing mass for Markov chains with mixing time 2 or 3.

Another metric that is extensively used in the study of Markov chains is spectral gap (Gallager, 1996; Levin et al., 2008). A chain with nonzero, constant spectral gap shares several properties of i.i.d sequences. However, through a counter example, we show how the GT estimator fails to converge to missing mass for Markov chains with a non-vanishing spectral gap. Hence, the success of the GT estimator for missing mass in the Markov case appears to require a new measure of closeness to i.i.d, and we study such a closeness property in this work.

We make two main contributions. Firstly, we study the Good-Turing (GT) estimator and characterise the classes of Markov chains or t.p.ms for which it converges to missing stationary mass. A large class of t.p.ms occurring in practical scenarios are likely to satisfy these characterisations.

Secondly, on the theoretical side, we characterise the minimax squared-error risk of estimating missing stationary mass over a class of rank-2 Markov t.p.ms with a spectral gap. To the best of our knowledge, this work presents the first minimax rate result for a Markov case.

## 3 Main Results

### 3.1 Convergence of GT estimator

We first provide a simplified expression for the bias of the GT estimator for missing stationary mass, using which convergence analysis becomes possible. We require the following notation.

For $x \in \mathcal{X}$, let $P^{\sim x}$ be a modified transition matrix equal to $P$ in all positions except the $x$-th column, which is set as the all-0 vector. So, under $P^{\sim x}$, the symbol $x$ is never observed. Let $P_{\downarrow x}^0 \triangleq P - P^{\sim x}$ be all-zero except for the $x$-th column, which is set as the $x$-th column of $P$. Let $\boldsymbol{\pi}^{\sim x}$ be equal to the vector $\boldsymbol{\pi}$ in all positions except the $x$-th entry, which is set as 0. Let $\mathbf{1}$ be the $|\mathcal{X}| \times 1$ vector with all entries as 1 and $\mathbf{e}_x$ be the $1 \times |\mathcal{X}|$ vector with the $x$-th entry as 1 and all other entries as zeros.

**Lemma 1.** *For a stationary Markov chain $X^n \sim Markov(P, \boldsymbol{\pi})$, the bias of the Good-Turing estimator can be expressed as follows:*

$$E[\widehat{M_0}^{GT}(X^n) - M_0(\boldsymbol{\pi}, X^n)] = \frac{1}{n} \sum_{x \in \mathcal{X}} \left[ \pi_x (\mathbf{e}_x - \boldsymbol{\pi}^{\sim x}) (P^{\sim x})^{(n-1)} \mathbf{1} \right.$$
$$\left. + \sum_{m=2}^{n} \boldsymbol{\pi}^{\sim x} (P^{\sim x})^{(m-2)} (P_{\downarrow x}^0 - \pi_x P^{\sim x}) (P^{\sim x})^{(n-m)} \mathbf{1} \right]. \tag{4}$$

*Proof.* Section 5. □

---

[1] In cases where missing mass is expected to vanish with $n$, relative error is more meaningful as shown in (Mossel & Ohannessian, 2019). However, in many interesting large alphabet scenarios, missing mass is non-vanishing for large $n$, and estimation is critical in the non-vanishing case. So, we consider additive error in this work.

For $X^n \sim$ i.i.d $\boldsymbol{\pi}$, the t.p.m $P = \mathbf{1} \, \boldsymbol{\pi}$, $P^{\sim x} = \mathbf{1} \, \boldsymbol{\pi}^{\sim x}$ and the bias of the GT estimator equals $\sum_{x \in \mathcal{X}} \pi_x^2 \, (1 - \pi_x)^{n-1} = O(1/n)$. Therefore, the Good-Turing estimator has a vanishing bias for the missing stationary mass in the i.i.d case and one might expect $\widehat{M}_0^{\mathrm{GT}}$ to perform similarly over Markov chains that are close to the i.i.d regime, say, in mixing time or spectral gap. In Section 3.3, we show through simulations that there exists a non-i.i.d Markov chain with mixing time as small as 2 (or 3), which is the closest non-i.i.d Markov chains can get to the i.i.d regime in terms of mixing time, for which the Good-Turing estimator does not converge to the missing stationary mass. A similar counterexample is shown for spectral gap as well. So we need a different notion of proximity to the i.i.d regime to extend the i.i.d result on the convergence of the GT estimator to the Markov regime.

In the following theorem, we present sufficient conditions on the t.p.m $P$ for the convergence of the GT estimator to $M_0(\boldsymbol{\pi}, X^n)$. These conditions require powers of $P^{\sim x}$ to be close to $\mathbf{1}\boldsymbol{\pi}^{\sim x}$, which is $P_{\mathrm{iid}}^{\sim x}$ of the corresponding i.i.d chain with t.p.m $P_{\mathrm{iid}} = \mathbf{1}\boldsymbol{\pi}$ and stationary distribution $\boldsymbol{\pi}$. In Section 3.3, we verify that these conditions are satisfied by randomly generated t.p.ms, empirical t.p.ms built using language corpora and the GT estimator has vanishing MSE for $M_0(\boldsymbol{\pi}, X^n)$ of the stationary Markov chains from these t.p.ms.

**Theorem 2.** *The absolute bias of the Good-Turing estimator $\widehat{M}_0^{GT}(X^n)$, for the missing stationary mass of a stationary Markov chain $X^n \sim Markov(P, \boldsymbol{\pi})$ is bounded as*

$$\left| E[\widehat{M}_0^{GT}(X^n) - M_0(\boldsymbol{\pi}, X^n)] \right| \leq \frac{2(2n_0 + 1)}{n} + [3\epsilon_{n_0,n} + \epsilon_{n_0,n}^2] + 2\frac{e^{-1}}{n-4} + a_0 \frac{e^{-1}}{c_1(n-2)}, \tag{5}$$

*where $n_0$, $\epsilon_{k,n}$, $a_0$ and $c_1$ (defined below) satisfy the following conditions:*

1. *there exist $n_0 = o(n)$, $\lambda_x \in [0,1]$ for $x \in \mathcal{X}$ and $\epsilon_{k,n} > 0$ such that for any $x \in \mathcal{X}$ and $k \geq n_0$,*

   (a) $(\lambda_x^{k-1} - \epsilon_{k,n}) \, \boldsymbol{\pi}^{\sim x} \leq \mathbf{e}_z \, (P^{\sim x})^k \leq (\lambda_x^{k-1} + \epsilon_{k,n}) \, \boldsymbol{\pi}^{\sim x}, \; z \in \mathcal{X}$,
   (b) $\lambda_x \leq 1 - c_1 \pi_x, \; c_1 \in [0,1]$,

2. *there exists $a_0 > 0$ such that $P_{xx} \leq a_0 \, \pi_x$ for all $x \in \mathcal{X}$.*

*The MSE of the GT estimator $\widehat{M}_0^{GT}(X^n)$ for $M_0(\boldsymbol{\pi}, X^n)$ of $X^n \sim Markov(P, \boldsymbol{\pi})$ is bounded as*

$$E[(\widehat{M}_0^{GT}(X^n) - M_0(\boldsymbol{\pi}, X^n))^2] \leq \frac{2}{n} \, (5n_0' + 2) \left[ 2 + \frac{3e^{-1}}{c_2(n-2)} \right] + \epsilon_{n_0,n} + 6\epsilon_{n_0',n}' + 4(\epsilon_{n_0',n}')^2 + (\epsilon_{n_0',n}')^3$$

$$+ \frac{1}{n-6} \left[ 2 + e^{-1} \left( 8e^{-1} + c_1^{-1} + c_2^{-1} \, (3 + 8a_0 e^{-1}) \right) \right] \tag{6}$$

*if, in addition to the above mentioned conditions, there exist $n_0' = o(n)$, $\{\lambda_{x,y} \in [0,1] : x, y \in \mathcal{X}, x \neq y\}$ and $\epsilon_{k,n}' > 0$ such that for any $x, y \in \mathcal{X}, x \neq y$ and $k \geq n_0'$,*

   (A) $(\lambda_{x,y}^{k-1} - \epsilon_{k,n}') \, \boldsymbol{\pi}^{\sim x,y} \leq \mathbf{e}_z \, (P^{\sim x,y})^k \leq (\lambda_{x,y}^{k-1} + \epsilon_{k,n}') \, \boldsymbol{\pi}^{\sim x,y}, \; z \in \mathcal{X}$,

   (B) $\lambda_{x,y} \leq 1 - c_2(\pi_x + \pi_y), \; c_2 \in [0,1]$,

*where the matrix $P^{\sim x,y}$ equals the t.p.m $P$ in all entries except in the $x$-th and $y$-th columns which are set to 0 and $\boldsymbol{\pi}^{\sim x,y}$ is the vector obtained by setting the $x$-th and $y$-th entries to 0 in $\boldsymbol{\pi}$.*

*Proof.* Section 6. $\qquad \square$

It is easy to retrieve the classical $O(1/n)$ bounds on the bias and MSE of the GT estimator for missing mass in $n$ i.i.d samples from Theorem 2 by checking that the conditions required for (5) and (6) are satisfied in the i.i.d case with $\lambda_x = 1 - \pi_x, \lambda_{x,y} = 1 - (\pi_x + \pi_y), \epsilon_{k,n} = \epsilon_{k,n}' = 0, c_1 = c_2 = a_0 = 1$ and $n_0 = n_0' = 1$.

The conditions on the t.p.m $P$ in the above theorem constrain the rows of $(P^{\sim x})^k$ to be similar to those of $(\mathbf{1} \, \boldsymbol{\pi}^{\sim x})^k = (1 - \pi_x)^{k-1} \, \mathbf{1} \, \boldsymbol{\pi}^{\sim x}$. This is reasonable and may be satisfied by many t.p.ms since the $k$-th

power of a non-negative matrix converges to the $k$-th power of its *Perron eigenvalue*, i.e. the eigenvalue with the largest magnitude, times the outer-product of the corresponding right and left eigenvectors and we expect such left and right eigenvectors of $P^{\sim x}$ to be close to those of $\mathbf{1}\boldsymbol{\pi}^{\sim x}$ i.e. $\boldsymbol{\pi}^{\sim x}$ and $\mathbf{1}$, and the Perron eigenvalue of $P^{\sim x}$ to be close to $1 - \pi_x$, the only non-zero eigenvalue of $\mathbf{1}\;\boldsymbol{\pi}^{\sim x}$. The conditions on $P^{\sim x,y}$ have similar interpretations.

The error term in the convergence of $(P^{\sim x})^k$, which we have denoted $\epsilon_{k,n}$, may, in general, be a function of $k$ and $n$ (note that entries of $P$ may be scaling with $n$ as well). The convergence of bias to zero depends on the convergence of $\epsilon_{n_0,n}$ to zero. This may be numerically verified for many interesting t.p.ms, as shown later. Similarly, the convergence of MSE to zero depends on the convergence of $\epsilon'_{n'_0,n}$ to zero.

The condition on $P_{xx}$ (self-loop probability) is necessary for successful convergence of the GT estimator as shown later by a counter example. This condition is also expected to be satisfied by a wide range of t.p.ms such as those arising from natural language text because a word seldom follows itself in writing.

The conditions of Theorem 2 need to be satisfied only for those letters $x$ that are most likely to contribute to the missing mass. For example, if $\pi_x \gg 1/n$, the letter $x$ will appear in $n$ samples with high probability and $P^{\sim x}$ need not satisfy the conditions. We have not explicitly stated such modifications for simplicity.

The conditions of Theorem 2 constrain a Markov t.p.m to be close to the i.i.d regime in an analytical way. Next, we shift to an algebraic view, and study missing mass estimation from Markov chains with t.p.ms that are close to the i.i.d regime in *rank*. Note that the i.i.d t.p.m has rank 1. As a natural next step, we consider t.p.ms with rank equal to 2.

### 3.2 Rank-2 Markov chains

Consider a Markov chain with a rank-2 t.p.m $P$, which we will call, loosely, as a rank-2 Markov chain. Since $P$ has all entries in $[0,1]$ with each row adding to 1 and since it has rank 2, the eigenvalues of $P$ will be 1, $\lambda_2$, 0, ..., 0, and $-1 \leq \lambda_2 \leq 1$ (by Perron-Frobenius theorem) (Pillai et al., 2005). The value of $\lambda_2$ determines several important properties of the chain. If $\lambda_2 = 1$, the chain is reducible. If $\lambda_2 = -1$, the chain is periodic with period 2. If $\lambda_2 = 0$, the chain is i.i.d For $-1 < \lambda_2 < 1$, the chain is irreducible and aperiodic. We define the spectral gap of a t.p.m $P$ as

$$\beta(P) \triangleq 1 - \lambda_2 \in [0, 2]. \tag{7}$$

In this section, we let $\mathcal{P}_{2,\beta}$ denote the family of rank-2 diagonalizable t.p.ms with spectral gap $\beta$.

### 3.2.1 Bias of GT estimator

The absolute bias of the GT estimator converges as $1/(n\beta^2) + 1/(n\beta^3)$ for rank-2 Markov chains with spectral gap $\beta$. This is shown in the next theorem.

**Theorem 3.** *For $P \in \mathcal{P}_{2,\beta}$ with stationary distribution $\boldsymbol{\pi}$ and $\beta \geq \left[30(\ln n)/(n-3)\right]^{1/2}$, there exists universal constants $c_1, c_2 > 0$, such that*

$$\left| E[\widehat{M}_0^{GT}(X^n) - M_0(\boldsymbol{\pi}, X^n)] \right| \; \leq \; \frac{c_1}{n\beta^2} + \frac{c_2}{n\beta^3} + \; O(1/n). \tag{8}$$

In comparison to Theorem 2, we see that the bias is explicitly bounded in terms of the spectral gap, which is an important parameter of the chain. For $\beta < 1$, we see that the bias of the GT estimator converges if $(1/(n\beta^3)) \to 0$, or $\beta$ is higher than $c/\sqrt[3]{n}$ for large $n$. So, even in cases where the spectral gap is asymptotically vanishing, the GT estimator converges if the rate of fall is not too rapid.

### 3.2.2 Minimax rate

For rank-2 chains, a much stronger result than convergence of bias can be shown. We next characterise the minimax rate $R_n^*(\mathcal{P}_{2,\beta})$ of the squared error risk of estimating $M_0(\boldsymbol{\pi}, X^n)$ of $X^n \sim \text{Markov}(P, \boldsymbol{\pi})$ for the class $\mathcal{P}_{2,\beta}$.

**Theorem 4.** *The minimax squared error risk $R_n^*(\mathcal{P}_{2,\beta})$ is bounded as follows:*

*1. For $\beta \geq ((160 \ln n)/(n-5))^{1/3}$,*

$$R_n^*(\mathcal{P}_{2,\beta}) \leq O(1/n\beta^5). \tag{9}$$

*2. For $n$ sufficiently large, there is a constant $c$ such that*

$$R_n^*(\mathcal{P}_\beta) \geq R_n^*(\mathcal{P}_{2,\beta}) \geq \frac{c}{n\beta}. \tag{10}$$

We get the upper bound in (9) by analyzing the worst-case MSE (over $\mathcal{P}_{2,\beta}$) of the GT estimator $\widehat{M}_0^{\mathrm{GT}}(X^n) = \phi_1(X^n)/n$ in estimating $M_0(\boldsymbol{\pi}, X^n)$ from $X^n \sim \mathrm{Markov}(P, \boldsymbol{\pi})$ with $P \in \mathcal{P}_{2,\beta}$. The simplifications in the upper bound are fairly involved and an outline of the proof is presented in Section 7. The complete proof is provided in the Appendix.

To get the lower bound in (10), we modify the Le Cam two point method. The usual Le Cam method for lower bounds on minimax risk (Yu, 1997) directly applies when the estimand does not depend on the samples, and is a parameter of the distribution alone. Using concentration properties, we extend the Le Cam two-point method to the case of estimating missing stationary mass $M_0(\boldsymbol{\pi}, X^n)$ that clearly depends on the samples $X^n$. By constructing two Markov chains (with t.p.ms in $\mathcal{P}_{2,\beta}$) close in distribution and separated in $M_0(\boldsymbol{\pi}, X^n)$, we get the lower bound in (10) on $R_n^*(\mathcal{P}_{2,\beta})$. Since $\mathcal{P}_{2,\beta}$ is contained in $\mathcal{P}_\beta$, the class of all Markov chains with spectral gap $\beta$, the same lower bound extends to $R_n^*(\mathcal{P}_\beta)$ as well. Specific details are provided in the Appendix.

Overall, we see that the minimax rate behaves as $1/n$ for rank-2 chains. This extends the previously known $1/n$ rate for i.i.d samples. The behaviour with respect to $\beta$ differs in the upper and lower bounds ($1/(n\beta^5)$ vs $1/(n\beta)$), and this gap could be closed in future work.

### 3.3 Synthetic and corpora-based illustrations

In this section, we present the results of our simulations studying the performance of the GT estimator in estimating the missing stationary mass of Markov sequences drawn using rank 2 t.p.ms, higher rank t.p.ms, randomly generated t.p.ms, and empirical t.p.ms built over natural language text.

#### 3.3.1 Rank-2 synthetic t.p.m

We consider a $K \times K$ rank-2 t.p.m with spectral gap $\beta$ formed by 4 $K/2 \times K/2$ blocks as follows:

$$\left[ \begin{array}{cccc|cccc} c_{K,\beta} & \cdots & c_{K,\beta} & c_{K,\beta} & \beta/2 & 0 & \cdots & 0 \\ \vdots & \ddots & \vdots & \vdots & \vdots & \vdots & \ddots & \vdots \\ c_{K,\beta} & \cdots & c_{K,\beta} & c_{K,\beta} & \beta/2 & 0 & \cdots & 0 \\ \hline 0 & \cdots & 0 & \beta/2 & c_{K,\beta} & c_{K,\beta} & \cdots & c_{K,\beta} \\ \vdots & \ddots & \vdots & \vdots & \vdots & \vdots & \ddots & \vdots \\ 0 & \cdots & 0 & \beta/2 & c_{K,\beta} & c_{K,\beta} & \cdots & c_{K,\beta} \end{array} \right],$$

where $c_{K,\beta} = (1 - \beta/2)(2/K)$. We generated stationary Markov sequences of lengths $n = 30, 60, 120, 240, 500, 1000$ from the rank$-2$ t.p.m specified above with $K = 1.2n$ and with $\beta = 1/n^{0.2}$, $\beta = 1/n$ and computed the missing mass $M_0$ and the GT estimate over 5000 trials. The mean values with standard deviation bars are shown in Fig. 1. For comparison, similar results are shown for i.i.d sequences from the same stationary distribution. As predicted by the theoretical results, the GT estimate converges in the rank-2 case for $\beta = 1/n^{0.2}$, while it is away by a constant value for $\beta = 1/n$. In the i.i.d case, there is convergence in all cases. We see that the variances in the rank-2 case are noticeably higher when compared to the i.i.d case.

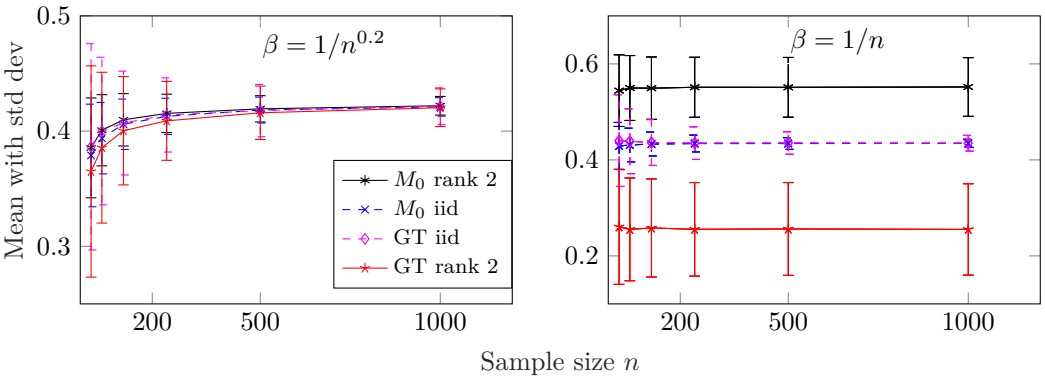

Figure 1: Missing mass and GT estimates for rank-2 chain and i.i.d sequence.

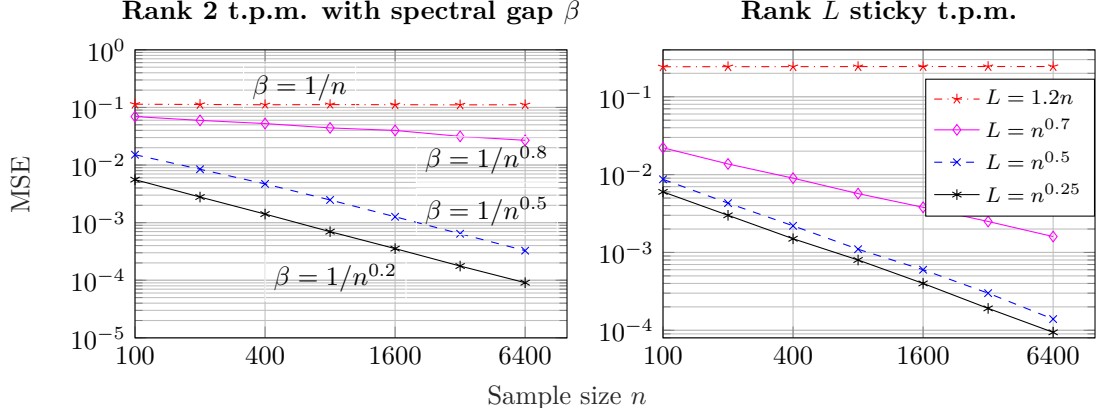

Figure 2: MSE of GT for a rank-2 chain with spectral gap $\beta$ and rank $L$ sticky t.p.m.

A plot of MSE of GT versus $n$ is shown in Fig. 2 for a larger range of values for $n$ for the rank-2 chain considered above. The parameters were chosen as $K = 1.2n, \beta = 1/n^{0.2}, 1/n^{0.5}, 1/n^{0.8}, 1/n$, and the MSE was averaged over 16000 trials for each $n$. We observe that the MSE falls with $n$ for $\beta$ higher than $1/n$, while it stays flat for $\beta = 1/n$. So, in this particular case, it appears that the actual MSE is as per the lower bound of Theorem 4 in terms of $\beta$.

### 3.3.2 Markov chains with rank of t.p.m above 2

The mean square error of the GT estimator for the missing stationary mass of a Markov chain with t.p.m rank greater than 2 is considered next for simulations. Through these simulations, we show the following:

- The rate of decay (with number of samples) of MSE of the GT estimator for the missing stationary mass of a Markov chain appears to vary with the rank of the t.p.m, possibly as one of the parameters.

- Rapid mixing of a Markov chain is not sufficient for convergence of GT estimator of missing mass.

- Large spectral gap of a Markov chain is not sufficient for convergence of GT estimator of missing mass.

We consider a $K \times K$ t.p.m $P$ with the $i$-th row as (recall that $\mathbf{e}_i$ is a vector with 1 at the $i$-th position and zero elsewhere)

$$P_i = \alpha \, \mathbf{e}_i + (1 - \alpha) \, \boldsymbol{p}, \text{ for } i = 1, 2, \ldots, L, \text{ and } P_i = \boldsymbol{p}, \text{ for } i = L + 1, \ldots, K,$$

where $\alpha \in [0,1]$ and $\boldsymbol{p} = \{p_x : x \in \mathcal{X}\}$ is a probability distribution on $\mathcal{X} = \{1, \ldots, K\}$. The stationary distribution $\boldsymbol{\pi}$ of this t.p.m is such that

$$\pi_i \; = \; \frac{1}{\tau} \; p_i, \text{ for } i = 1, 2, \ldots, L, \text{ and } \pi_i \; = \; \frac{1 - \tau}{\tau} \; p_i, \text{ for } i = L+1, \ldots, K,$$

where $\tau = 1 - \alpha \left( \sum_{i=L+1}^{K} p_i \right)$.

For $\alpha > 0$ and $L = K$, $\boldsymbol{\pi} = \boldsymbol{p}$ and we refer to the above t.p.m as a (geometrically) *sticky* t.p.m or a sticky Markov chain. For $\alpha > 0$ and $L < K$, we refer to the above t.p.m as a *partially sticky* t.p.m. For $\alpha = 0$, we retrieve the i.i.d chain. Note that the rank of this t.p.m is $L+1$ for $L < K$ and $K$, i.e. full rank, for $L = K$.

We generated stationary Markov sequences of lengths $n = 100, 200, 400, 800, 1600, 3200, 6400$ from the t.p.m specified above with $K = 1.2n$, $L = n^{0.25}, n^{0.5}, n^{0.7}, 1.2n$, $\alpha = 0.5$ and $\boldsymbol{p}$ as the uniform distribution.

The scaling of the alphabet size $K$ (and, as a result, the parameters of the t.p.m) with the sample size $n$ models the large alphabet nature of applications like language text where the alphabet size is greater or of the same order as the data size. The missing mass estimation problem is non-trivial in cases such as these since, for example, the (expected value of) missing mass in $n$ samples drawn from a uniform distribution on $cn$ letters, $c$ being a constant, converges to $e^{-1/c}$ as $n \to \infty$, whereas in cases with the alphabet size or parameters of t.p.m independent of $n$, the (expected value of) missing mass in $n$ samples of a sequence generated from such t.p.ms converges to 0 as $n$ increases. Note that the MSE of the GT estimator for the missing mass in $n$ i.i.d samples (i.e. rank-1 t.p.m) of a uniform distribution on $cn$ letters decays as $1/n$ (Rajaraman et al., 2017).

A plot of the MSE (averaged over 16000 trials) of GT against $n$, for these choices of $L$, is shown in Fig 2. From the plot, we observe that the MSE of the GT estimator, for missing stationary mass of a Markov chain with the above specified t.p.m, increases with the rank of the t.p.m and the rate of decay of the MSE depends on the rank.

Further, the MSE is non-vanishing (with $n$) when the t.p.m is of full rank which is a geometrically sticky t.p.m in this case. Note that the geometrically sticky Markov chain considered here with $\alpha = 0.5$ has mixing times of 2 and 3 for total-variation distances of $1/4$ and $1/8$ to the stationary distribution $\boldsymbol{\pi}$. Further, the spectral gap is $1 - \alpha = 0.5$. Therefore, even in the regime of mixing times close to i.i.d (the mixing time of an i.i.d chain is 1) and constant spectral gap, there exist Markov chains for which the GT estimator does not converge to $M_0(\boldsymbol{\pi}, X^n)$.

The geometrically sticky chain violates Condition 2 of Theorem 2, i.e. $P_{xx} = \alpha + (1 - \alpha)\pi_x \leq a_0 \; \pi_x$ is not satisfied for $\alpha \gg \pi_x$. The authors of (Chandra et al., 2022) suggest a scaling of the GT estimator, which converges to missing mass in this case.

### 3.3.3 T.p.ms generated at random and from corpora

Analytical characterisation of non-i.i.d t.p.ms that satisfy the conditions in Theorem 2 is a challenging problem. While there are counterexamples such as the sticky channels, the GT estimator appears to work in practice for several text-based corpora, which behave like Markov chains. To study this phenomenon under the setting of Theorem 2, we construct some random t.p.ms and some t.p.ms from text corpora and verify conditions in Theorem 2 numerically.

We consider a class of randomly generated t.p.ms, and two classes of t.p.ms from text corpora.

- $P_{\text{unif-gen}}$: Each entry is first drawn i.i.d from the uniform distribution over $[0,1]$ and each row is then scaled to make the row sum equal to 1.

- NYT: The New York Times (NYT) corpus [2] consists of randomly collected articles from the front pages of New York Times from the years 2017 and 2018. To build an empirical t.p.m, we consider an

---

[2]available under CC0:Public domain license at https://www.kaggle.com/datasets/mathurinache/10700-articles-from-new-york-times

article in the NYT corpus and set the empirical transition probability (from word $w_1$ to $w_2$) $P_{w_1,w_2}$ as $N_{w_1,w_2}/N_{w_1}$, where $N_{w_1,w_2}$ is the number of times the word $w_2$ follows $w_1$ in the article and $N_w$ is the number of occurences of the word $w$ in the article [3]. The empirical probability $\pi_w$ is set to $N_w/(\text{total wordcount})$ of the article.

- GE: The novel *Great Expectations* (GE) by Charles Dickens is available under the project Gutenberg (https://www.gutenberg.org/). To construct a t.p.m, we consider a chapter from the novel and repeat the same process as NYT above.

We specifically use two $P_{\text{unif-gen}}$ t.p.ms with support sizes $K = 1250, 1.2n$ and three empirical t.p.ms, each built using a chapter of the novel *Great Expectations* and five empirical t.p.ms, each built using an article with more than 1600 words from the NYT corpus. To verify Condition 1 in Theorem 2, we use the Perron eigenvalue $\lambda_{1,x}$, the eigenvalue of $P^{\sim x}$ with the largest magnitude. We consider the difference between the ratio $(P^{\sim x})^k_{yz}/\pi_z$ and $(\lambda_{1,x})^{k-1}$ and denote by $\epsilon_{k,n,x}$, the maximum of the absolute value of this difference over all the entries of $(P^{\sim x})^k$, except the entries of the $x$-th column which are 0. $\epsilon_{k,n}$ is the maximum of $\epsilon_{k,n,x}$ over all $x$. Fig 3 shows two scatter plots, for $k = 16$ and $64$, of $\epsilon_{k,n,x}$ against $1 - \pi_x$ for a $P_{\text{unif-gen}}$ t.p.m with $K = 1.2n$ and $n = 100, 200$ and $400$. From these plots, we see that $\epsilon_{k,n,x}$ falls with both $k$ and $n$. Fig 3 also shows a scatter plot of $\lambda_{1,x}$ against $1 - \pi_x$ for the same t.p.m with $n = 100, 200$ and $400$, which indicates a linear relation between $\lambda_{1,x}$ and $1 - \pi_x$. From the plots in Fig 3, we see that Condition 1 in Theorem 2 is satisfied by this t.p.m with $\lambda_x$ as $\lambda_{1,x}$ as $\epsilon_{k,n}$ decreases with $n$.

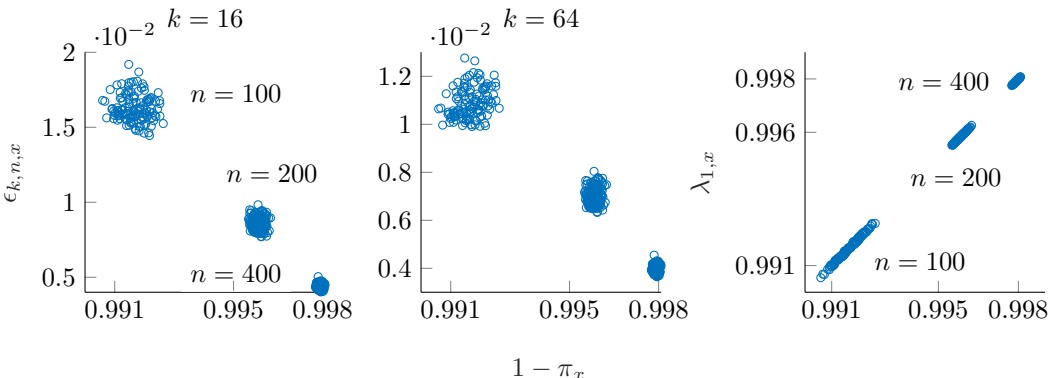

Figure 3: $\epsilon_{k,n,x}$ vs $1 - \pi_x$ and $\lambda_{1,x}$ vs $1 - \pi_x$ for $P_{\text{unif-gen}}$ t.p.ms with $K = 1.2n$

Figure 4 shows scatter plots of $1 - \lambda_{1,x}$ against the stationary probability $\pi_x$ for the remaining three t.p.ms. All the plots show a linear upper bound relation between $1 - \lambda_{1,x}$ and $\pi_x$ closely matching the Condition 1(b) in Theorem 2. In addition, these t.p.ms satisfy Condition 1(a) in Theorem 2 as $\epsilon_{n_0,n}$ is negligible for a sufficiently high $n_0$.

For the randomly generated t.p.ms $P_{\text{unif-gen}}$ with $K = 1250$ and $K = 1.2n$, $P_{xx}$ was found to be less than $10\pi_x$, for any $x$. For the empirical t.p.ms built from language text, $P_{xx} = 0$ for any $x$. Therefore, the t.p.ms under consideration also satisfy Condition 2 in Theorem 2. Since $\epsilon_{n_0,n}$ falls with $n$ for the $P_{\text{unif-gen}}$ t.p.m with $K = 1.2n$ and is negligible for a suitable choice of $n_0$ for the other t.p.ms, Theorem 2 implies that the GT estimator should converge to the missing stationary mass of Markov chains from these t.p.ms.

Figure 5 plots the MSE of the GT estimator for $M_0(\boldsymbol{\pi}, X^n)$ of a stationary Markov chain $X^n$ generated using the t.p.ms $P_{\text{unif-gen}}$ with $K = 1250, 1.2n$, the three empirical t.p.ms from GE and the five empirical t.p.ms from NYT corpora, for $n = 100, 200, 400, 800, 1600, 3200, 6400$ and averaged over 16000 trials. The curves for GE and NYT correspond to MSE averaged over the three t.p.ms from *Great Expectations* and the five t.p.ms from the NYT corpus. We observe that MSE falls for all of these t.p.ms.

---

[3]We change all the words in the article to lower case, lemmatize the words and ignore punctuations.

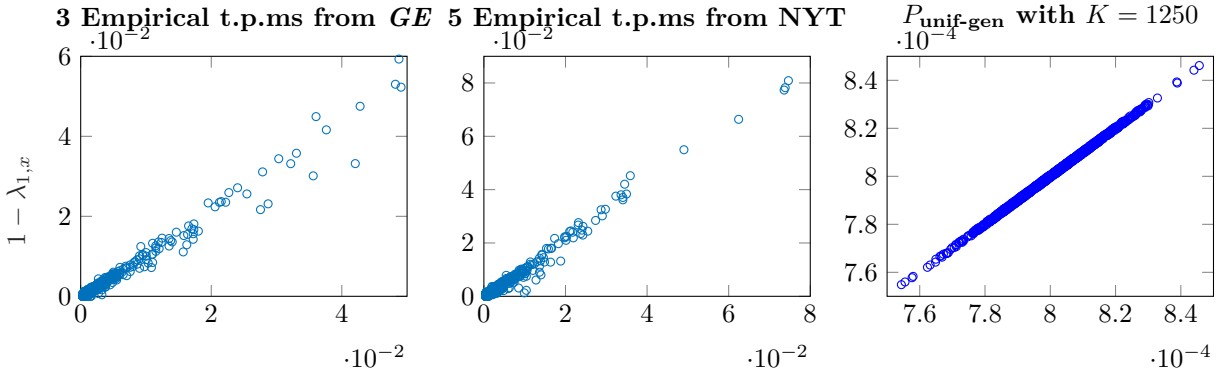

Figure 4: $1 - \lambda_{1,x}$ vs $\pi_x$ for randomly generated t.p.m and t.p.ms from corpora.

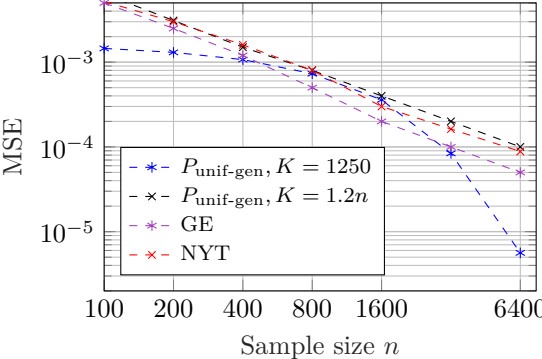

Figure 5: MSE of GT for randomly generated t.p.ms and t.p.ms from corpora.

## 4 Conclusion and Future Directions

In conclusion, our study of the Good-Turing (GT) estimator for missing stationary mass of a Markov chain with t.p.m $P$ and stationary probability $\boldsymbol{\pi}$ (with support size assumed to be unknown) indicates that convergence of GT depends on relationships between the spectrum of $P^{\sim x}$ ($P$ with $x$-th column zeroed out) and $\pi_x$ to be satisfied uniformly for all $x$ that contribute to missing mass. We derive specific sufficient conditions for convergence of absolute bias in terms of the nature of convergence of powers of $P^{\sim x}$ with respect to powers of $P$. These conditions are verified numerically for t.p.ms derived from text corpora, which supports the success of GT in practice. Analytical understanding of relationships between the spectrum of $P^{\sim x}$ and $\pi_x$ for arbitrary t.p.ms is a topic for future study. In the case when $P$ has rank 2 with a spectral gap of $\beta$, we derive lower ($c/(n\beta)$) and upper ($c'/(n\beta^5)$) bounds on the minimax squared-error risk. The bounds extend the $1/n$ minimax rate result from the i.i.d case to rank-2 Markov. Characterizing the exact dependence on $\beta$ is a topic for future work. Through our simulations on high-rank t.p.ms, we see that the rate of decay (with $n$) of the MSE of the GT estimator for the missing stationary mass might vary with rank of the t.p.m.

## 5 Proof of Lemma 1

To prove (4), we begin with the expressions for the expected values of $M_0(\boldsymbol{\pi}, X^n)$ and $\widehat{M}_0^{\mathrm{GT}}$ using $\boldsymbol{\pi}^{\sim x}, P^{\sim x}$ and $P_{\downarrow x}^0$.

Since $M_0(\boldsymbol{\pi}, X^n) = \sum_{x \in \mathcal{X}} \pi_x \, I(N_x(X^n) = 0)$, we have

$$E[M_0(\boldsymbol{\pi}, X^n)] \;=\; \sum_{x \in \mathcal{X}} \pi_x \, \Pr(N_x(X^n) = 0) \;=\; \sum_{x \in \mathcal{X}} \pi_x \, \boldsymbol{\pi}^{\sim x} \, (P^{\sim x})^{n-1} \, \mathbf{1}. \tag{11}$$

Since $\widehat{M}_0^{\mathrm{GT}}(X^n) = \phi_1(X^n)/n = \frac{1}{n} \sum_{x \in \mathcal{X}} I(N_x(X^n) = 1)$, we have

$$E[\widehat{M}_0^{\mathrm{GT}}(X^n)] \;=\; \frac{1}{n} \sum_{x \in \mathcal{X}} \Pr(N_x(X^n) = 1) \;=\; \frac{1}{n} \sum_{x \in \mathcal{X}} \sum_{m=1}^{n} \Pr(X_m = x; X_l \neq x, l \neq m, 1 \leq l \leq n)$$

$$= \frac{1}{n} \sum_{x \in \mathcal{X}} \left[ \pi_x \, \mathbf{e}_x \, (P^{\sim x})^{(n-1)} \, \mathbf{1} + \sum_{m=2}^{n} \boldsymbol{\pi}^{\sim x} \, (P^{\sim x})^{m-2} \, P_{\downarrow x}^0 \, (P^{\sim x})^{n-m} \, \mathbf{1} \right]. \tag{12}$$

Taking the difference (12) and (11), we get (4). This completes the proof of Lemma 1.

## 6   Proof of Theorem 2

In this section, we provide a proof for (5) in Theorem 2. The proof for (6) is similar. To prove the bound on the bias of $\widehat{M}_0^{\mathrm{GT}}$ in (5), we first bound the expectation of $\widehat{M}_0^{\mathrm{GT}}$. Let $X_{\sim m}^n \triangleq (X_1, \ldots, X_{m-1}, X_{m+1}, \ldots, X_n)$, the samples $X_1 \ldots X_n$ except $X_m$, for $m = 1, \ldots, n$.

$$E[\widehat{M}_0^{\mathrm{GT}}(X^n)] \;=\; \frac{1}{n} \sum_{x \in \mathcal{X}} \Pr(N_x(X^n) = 1) \;=\; \frac{1}{n} \sum_{x \in \mathcal{X}} \sum_{m=1}^{n} \Pr(X_m = x, N_x(X_{\sim m}^n) = 0)$$

$$= \frac{1}{n} \sum_{x \in \mathcal{X}} \left[ \sum_{\substack{m=1,\ldots,n_0+1 \\ n-n_0+1,\ldots,n}} \Pr(X_m = x, N_x(X_{\sim m}^n) = 0) + \sum_{m=n_0+2}^{n-n_0} \Pr(X_m = x, N_x(X_{\sim m}^n) = 0) \right]$$

$$\overset{(a)}{\leq} \frac{1}{n} \sum_{x \in \mathcal{X}} \left[ \sum_{\substack{m=1,\ldots,n_0+1 \\ n-n_0+1,\ldots,n}} \Pr(X_m = x) + \sum_{m=n_0+2}^{n-n_0} \Pr(X_m = x, N_x(X_{\sim m}^n) = 0) \right]$$

$$= \frac{1}{n} \sum_{x \in \mathcal{X}} \left[ (2n_0 + 1) \, \pi_x + \sum_{m=n_0+2}^{n-n_0} \boldsymbol{\pi}^{\sim x} \, (P^{\sim x})^{m-2} \, P_{\downarrow x}^0 \, (P^{\sim x})^{n-m} \, \mathbf{1} \right]$$

$$\overset{(b)}{\leq} \frac{1}{n} \sum_{x \in \mathcal{X}} \left[ (2n_0 + 1) \, \pi_x + \sum_{m=n_0+2}^{n-n_0} \boldsymbol{\pi}^{\sim x} \, (\lambda_x^{m-3} + \epsilon_{m-2,n}) \, \mathbf{1} \, \boldsymbol{\pi}^{\sim x} \, P_{\downarrow x}^0 \, (\lambda_x^{n-m-1} + \epsilon_{n-m,n}) \, \mathbf{1} \, \boldsymbol{\pi}^{\sim x} \, \mathbf{1} \right]$$

$$\overset{(c)}{=} \frac{1}{n} \sum_{x \in \mathcal{X}} \Bigg[ (2n_0 + 1) \, \pi_x + \sum_{m=n_0+2}^{n-n_0} \pi_x \, (1 - P_{xx}) \, (1 - \pi_x)^2 \, (\lambda_x^{n-4} + \epsilon_{n-m,n} \, \lambda_x^{m-3} + \epsilon_{m-2,n} \, \lambda_x^{n-m-1}$$

$$+ \epsilon_{m-2,n} \, \epsilon_{n-m,n}) \Bigg]$$

$$\overset{(d)}{\leq} \frac{2n_0 + 1}{n} + \left( 1 - \frac{2n_0 + 1}{n} \right) \sum_{x \in \mathcal{X}} \left[ \pi_x \, (1 - P_{xx}) \, (1 - \pi_x)^2 \, (\lambda_x^{n-4} + 2\epsilon_{n_0,n} + \epsilon_{n_0,n}^2) \right]$$

$$\overset{(e)}{\leq} \frac{2(2n_0 + 1)}{n} + \left( 1 - \frac{2n_0 + 1}{n} \right) [2\epsilon_{n_0,n} + \epsilon_{n_0,n}^2] + \sum_{x \in \mathcal{X}} \pi_x \, (1 - P_{xx}) \, (1 - \pi_x)^2 \, \lambda_x^{n-4}, \tag{13}$$

where we get $(a)$ by using $\Pr(X_m = x; X_l \neq x, l \neq m, 1 \leq l \leq n) \;\leq\; \Pr(X_m = x) = \pi_x$, $(b)$ by using Condition 1(a) of Theorem 2, $(c)$ by using $\boldsymbol{\pi}^{\sim x} \, \mathbf{1} = 1 - \pi_x$ and $\boldsymbol{\pi}^{\sim x} \, P_{\downarrow x}^0 \, \mathbf{1} = \pi_x \, (1 - P_{xx})$, $(d)$ by using $\sum_{x \in \mathcal{X}} \pi_x = 1$, $|\lambda_x| \leq 1$ and $\epsilon_{n-m,n} > \epsilon_{n_0,n}$, $\epsilon_{m-2,n} > \epsilon_{n_0,n}$ for $m = n_0 + 2, \ldots, n - n_0$ and $(e)$ by using $\sum_{x \in \mathcal{X}} \pi_x \, (1 - P_{xx}) \, (1 - \pi_x)^2 \, \lambda_x^{n-4} \leq \sum_{x \in \mathcal{X}} \pi_x \, (1 - P_{xx}) \, (1 - \pi_x)^2 \leq \sum_{x \in \mathcal{X}} \pi_x = 1$.

Using a similar method, we lower bound the expectation of $M_0(\boldsymbol{\pi}, X^n)$ as

$$
E[M_0(\boldsymbol{\pi}, X^n)] = \sum_{x \in \mathcal{X}} \pi_x \, \boldsymbol{\pi}^{\sim x} \, (P^{\sim x})^{n-1} \, \mathbf{1} \overset{(a)}{\geq} \sum_{x \in \mathcal{X}} \pi_x \, \boldsymbol{\pi}^{\sim x} \, (\lambda_x^{n-2} - \epsilon_{n-1,n}) \, \mathbf{1} \, \boldsymbol{\pi}^{\sim x} \, \mathbf{1}
$$
$$
= \sum_{x \in \mathcal{X}} \pi_x \, (1 - \pi_x)^2 \, (\lambda_x^{n-2} - \epsilon_{n-1,n}), \tag{14}
$$

where we get $(a)$ by using Condition 1(a) of Theorem 2. Using (13) and (14), we get

$$
E[\widehat{M}_0^{\mathrm{GT}}(X^n)] - E[M_0(\boldsymbol{\pi}, X^n)] \overset{(a)}{\leq} \frac{2(2n_0 + 1)}{n} + [3\epsilon_{n_0,n} + \epsilon_{n_0,n}^2] + \sum_{x \in \mathcal{X}} \pi_x \, (1 - \pi_x)^2 \, \lambda_x^{n-4} \left[ 1 - P_{xx} - \lambda_x^2 \right]
$$
$$
\overset{(b)}{\leq} \frac{2(2n_0 + 1)}{n} + [3\epsilon_{n_0,n} + \epsilon_{n_0,n}^2] + \sum_{x \in \mathcal{X}} \pi_x (1 - \pi_x)^2 \lambda_x^{n-4} \left[ 2(1 - \lambda_x) + a_0 \pi_x \right]
$$
$$
\overset{(c)}{\leq} \frac{2(2n_0 + 1)}{n} + [3\epsilon_{n_0,n} + \epsilon_{n_0,n}^2] + 2\frac{e^{-1}}{n - 4} + a_0 \frac{e^{-1}}{c_1(n - 2)} \tag{15}
$$

where we get $(a)$ by using $\sum_{x \in \mathcal{X}} \pi_x \, (1 - \pi_x)^2 \, \epsilon_{n-1,n} \leq \epsilon_{n-1,n} \sum_{x \in \mathcal{X}} \pi_x \leq \epsilon_{n-1,n} \leq \epsilon_{n_0,n}$, $(b)$ by using $1 + \lambda_x \leq 2$, $P_{xx} \leq a_0 \, \pi_x$ and $(c)$ by using $\lambda_x \leq 1 - c_1\pi_x$ from Condition 1 in Theorem 2 along with $\max_{t \in [0,1]} t \, (1 - t)^n \leq e^{-1}/n$.

Using a similar method, we can show that

$$
E[M_0(\boldsymbol{\pi}, X^n)] - E[\widehat{M}_0^{\mathrm{GT}}(X^n)] \leq \frac{2(2n_0 + 1)}{n} + [3\epsilon_{n_0,n} + \epsilon_{n_0,n}^2] + 2\frac{e^{-1}}{n - 4} + a_0 \frac{e^{-1}}{c_1(n - 2)}.
$$

This completes the proof of (5) in Theorem 2.

## 7  Proof of Theorems 3 and 4

A proof of Theorem 3 is given in Section A of the Appendix. In the proof,

- we split the alphabet $\mathcal{X}$ into three sets: letters with $\pi_x = P_{xx}$, which we refer to as *iid-like* letters, *infrequent* letters with $\pi_x \neq P_{xx}$, *frequent* letters with $\pi_x \neq P_{xx}$ and bound the letter-wise bias of the GT estimator $\Gamma_x \triangleq \frac{1}{n} \Pr(N_x(X^n) = 1) - \pi_x \Pr(N_x(X^n) = 0)$, i.e. $E[\widehat{M}_0^{\mathrm{GT}}(X^n) - M_0(\boldsymbol{\pi}, X^n)] = \sum_{x \in \mathcal{X}} \Gamma_x$, over these sets. The *infrequent* letters contribute the most to missing mass, in expectation.

- For the letters with $\pi_x = P_{xx}$, $\Pr(N_x(X^n) = 0) = (1 - \pi_x)^n$ and $\Pr(N_x(X^n) = 1) = n \, \pi_x \, (1 - \pi_x)^{n-1}$ and the sum of $|\Gamma_x|$ over all the *iid-like* letters is bounded as $O(1/n)$.

- We bound the sum of $|\Gamma_x|$ over all *infrequent* letters (with $\pi_x \neq P_{xx}$) by using the eigenvalue decomposition of $P^{\sim x}$ in (4) along with bounds of the form $\lambda_{1,x} \leq 1 - c_0\pi_x$, $c_0$ being a function of the spectral gap $\beta$, on the Perron eigenvalue $\lambda_{1,x}$ of $P^{\sim x}$.

- To bound the sum of $|\Gamma_x|$ (over all *frequent* letters) with $\pi_x \neq P_{xx}$, we bound $\Pr(N_x(X^n) = 0)$ and $\Pr(N_x(X^n) = 1)$ using the eigenvalue decomposition of $P^{\sim x}$ along with bounds of the form $\lambda_{1,x} \leq 1 - c_0\pi_x$ and use these to upper bound the absolute value of $\Gamma_x$.

The upper bound on the bias of the GT estimator in (8) is obtained by combining the bounds on the sum of $|\Gamma_x|$ over these three sets and choosing a suitable threshold on stationary probability to split the alphabet $\mathcal{X}$ into frequent and infrequent letters.

The proof of the upper bound in Theorem 4 also uses ideas similar to the above. See Section B in the Appendix.

The lower bound in Theorem 4 is proved by an extension of the standard Le Cam method to estimation of the missing mass random variable. See Section C in the Appendix.

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

## A  Proof of Theorem 3

Consider a rank-2, $K \times K$ diagonalizable t.p.m $P$ with stationary distribution $\boldsymbol{\pi} = [\pi_1 \ \cdots \ \pi_K]$ and spectral gap $\beta$. By the standard eigenvector decomposition for the rank-2 matrix $P$, there exist vectors $u = [u_1 \ \cdots \ u_K]$ and $v = [v_1 \ \cdots \ v_K]^T$ satisfying the decomposition

$$P = RDS \text{ with } SR = \begin{bmatrix} 1 & 0 \\ 0 & 1 \end{bmatrix},$$

where $R = \begin{bmatrix} \mathbf{1} & v \end{bmatrix}$, $D = \begin{bmatrix} 1 & 0 \\ 0 & 1 - \beta \end{bmatrix}$ and $S = \begin{bmatrix} \boldsymbol{\pi} \\ u \end{bmatrix}$. Since $P$ is a t.p.m, we have, for $1 \leq i, j \leq K$,

$$0 \leq P_{ij} = \pi_j + \overline{\beta} v_i u_j \leq 1, \tag{16}$$

where the notation $\overline{a} \triangleq 1 - a$.

Since $P = RDS$, we have $P^{\sim x} = RDS^{\sim x}$ where $S^{\sim x}$ is obtained by setting the $x$-th column in $S$ to zeros. Now, the matrix $P^{\sim x}$, when diagonalizable, can be written as

$$P^{\sim x} = \sum_{i=1}^{2} \lambda_{i,x} \ v_i^{\sim x} \ u_i^{\sim x}, \tag{17}$$

with

$$\text{eigenvalues} \quad \lambda_{i,x} \triangleq 0.5\big(\overline{\pi}_x + \overline{\beta}(1 - v_x u_x) + (-1)^{i+1}\Delta_x\big), \tag{18}$$

$$\text{right eigenvectors} \quad v_i^{\sim x} = \mathbf{1} + (1/2\pi_x v_x) \left[ s_x + (-1)^i \Delta_x \right] v, \tag{19}$$

$$\text{left eigenvectors} \quad u_i^{\sim x} = (1/\lambda_{i,x}\Delta_x) \left( (1/2) \left[ \Delta_x + (-1)^{i-1} s_x \right] \boldsymbol{\pi}^{\sim x} + (-1)^i \overline{\beta}\pi_x v_x \ u^{\sim x} \right), \tag{20}$$

for $i = 1, 2$, where $\Delta_x^2 \triangleq s_x^2 + 4\overline{\beta}\pi_x v_x u_x$ and $s_x \triangleq \beta - \pi_x + \overline{\beta}v_x u_x$. We use $u^{\sim x}$ to denote the vector $u$ with $x$-th entry set to 0. Note $\boldsymbol{\pi}v_i^{\sim x} = 1$ for $i = 1, 2$, since $\boldsymbol{\pi}v = 0$. $P^{\sim x}$, in the rank-2 case, is not diagonalizable when both its non-zero eigenvalues equal $1 - \pi_x$ with only one non-trivial eigenvector and this case is handled separately later (refer lemma 8).

The right and left eigenvectors of $P^{\sim x}$ are expressed in terms of $\mathbf{1}$ and $\boldsymbol{\pi}$ (or $\boldsymbol{\pi}^{\sim x}$), the right and left eigenvectors of $P$. However, the difference terms involve the eigenvalues and eigenvector coordinates, which need to be carefully bounded.

The following lemma contains important relationships between $\lambda_{i,x}$, $\pi_x$ and $\beta$.

**Lemma 5.** *1.*

$$\Delta_x \in [-(\beta(1 - \pi_x) + P_{xx}), \ (\beta(1 - \pi_x) + P_{xx})], \ \text{if } \beta \in [0, 1] \tag{21}$$

*2.*

$$\Delta_x \in [-(\beta + (1 - \beta)v_x u_x), \ (\beta + (1 - \beta)v_x u_x)], \ \text{if } \beta \in [1, 2] \tag{22}$$

*3.*

$$|\lambda_{i,x}| \ \leq \ 1 - (c_\beta/2)\pi_x, \ i = 1, 2, \tag{23}$$

*where $c_\beta = \beta$ for $\beta \in (0, 1]$, $c_\beta = 1$ for $\beta \in [1, 2]$.*

*4.*

$$|\lambda_{i,x}| \ \leq \ 1 - \frac{\beta^2}{2(\beta + 2)}\pi_x, \ i = 1, 2. \tag{24}$$

*Proof.* See Section A.4. $\qquad\qquad\qquad\qquad\qquad\qquad\qquad\qquad\qquad\qquad\qquad\qquad\square$

In the eigenvalue decomposition of $P$, let $\psi_{wz} \triangleq \overline{\beta}v_w u_z$, for $w, z \in \mathcal{X}$. The following lemma bounds summation terms that typically occur in the analysis.

**Lemma 6.** *For $x, y \in \mathcal{X}$,*

$$\sum_{x \in \mathcal{X}} (\pi_x)^a \ |\psi_{xx}^b| \ |\psi_{yx}^c| \ \leq \ 3, \ \text{for } a, b, c \in \{0, 1, 2, 3, \ldots\} \text{ and } a + b + c \geq 1. \tag{25}$$

*Proof.* See Section A.4. $\qquad\qquad\qquad\qquad\qquad\qquad\qquad\qquad\qquad\qquad\qquad\qquad\square$

Let $\Gamma_x \triangleq \frac{1}{n} \Pr(N_x(X^n) = 1) - \pi_x \Pr(N_x(X^n) = 0)$ be the letter-wise bias of the GT estimator i.e. $E[\widehat{M}_0^{\text{GT}}(X^n) - M_0(\boldsymbol{\pi}, X^n)] = \sum_{x \in \mathcal{X}} \Gamma_x$. To bound $\Gamma_x$, we divide the alphabet $\mathcal{X}$ into three sets,

$$A_0 \triangleq \{x \in \mathcal{X} : \pi_x = P_{xx}\}, \text{ (iid-like)}$$

$$A(\delta) \triangleq \{x \in \mathcal{X} \setminus A_0 : \pi_x < \delta\}, \text{ (infrequent)}$$

$$\overline{A}(\delta) \triangleq \{x \in \mathcal{X} \setminus A_0 : \pi_x \geq \delta\}, \text{ (frequent)}$$

with $0 \leq \delta < \beta/5$. For $\delta = O((\ln n)/n)$, the letters in the set $A(\delta)$ are less likely to occur in $X^n$, than the letters in $\overline{A}(\delta)$, and contribute more to the missing mass $M_0(\boldsymbol{\pi}, X^n)$.

## A.1 Case 1: Infrequent letters

Using (17) and the ensuing expressions in (4), we obtain

$$\Gamma_x = \left[ \sum_{i=1}^{2} (\lambda_{i,x})^{n-1} \ (u_i^{\sim x}\mathbf{1}) \ (u_i^{\sim x}P_{\downarrow x}^0 v_i^{\sim x} - \pi_x\lambda_{i,x}) \right.$$
$$\left. + (1/n) \ \Delta_x^{-1} \ [(\lambda_{1,x})^n - (\lambda_{2,x})^n] \sum_{i,j\in[2]:\ i\neq j} (u_j^{\sim x}\mathbf{1}) \ (u_i^{\sim x}P_{\downarrow x}^0 v_j^{\sim x}) \right]. \tag{26}$$

For $x \in A(\delta)$, we will show by careful analysis that $\sum_{x\in A(\delta)} |\Gamma_x|$ is bounded. The following lemma bounds the absolute value of the factors multiplying $\lambda_{i,x}$ powers in the above expression for $\Gamma_x$ when $x \notin A_0$.

**Lemma 7.** *For $x \in \mathcal{X} \setminus A_0$,*

*1. For $i = 1, 2$,*

$$\left| (u_i^{\sim x}\mathbf{1}) \ (u_i^{\sim x} \ P_{\downarrow x}^0 \ v_i^{\sim x} - \pi_x\lambda_{i,x}) \right| \ \leq \ \Delta_x^{-2} \left[ 4 \ (|\psi_{xx}| + \pi_x) \ \pi_x + 2 \ \pi_x \ \lambda_{1,x} \ (1 - \lambda_{1,x}) \right]. \tag{27}$$

*2. For $i, j$ in $\{1, 2\}$ with $i \neq j$,*

$$\left| (u_i^{\sim x} \ \mathbf{1}) \ (u_j^{\sim x} \ P_{\downarrow x}^0 \ v_i^{\sim x}) \right| \ \leq \ 6 \ \Delta_x^{-2} \ \pi_x. \tag{28}$$

*Proof.* To prove Lemma 7, we use the expressions for $v_i^{\sim x}$ and $u_i^{\sim x}$ in (19) and (20) to get

1.
$$u_i^{\sim x} \ \mathbf{1} = \frac{1}{2}\left[ 1 + (-1)^{i-1} \frac{s_x}{\Delta_x} \right], i = 1, 2. \tag{29}$$

2.
$$u_i^{\sim x} \ P_{\downarrow x}^0 \ v_i^{\sim x} - \pi_x\lambda_{i,x} = \frac{1}{\Delta_x} \left[ \frac{1}{2}(\Delta_x + (-1)^i s_x)(\overline{\beta} - \lambda_{i,x}) \right.$$
$$\left. + (-1)^{i-1}\pi_x \ [\lambda_{i,x}(1 - \lambda_{i,x}) - \overline{\beta}(\pi_x + v_x u_x)] \right], i = 1, 2. \tag{30}$$

3.
$$u_1^{\sim x} \ P_{\downarrow x}^0 \ v_2^{\sim x} = \frac{1}{\Delta_x} \left[ \lambda_{1,x} - \overline{\beta} \right] \left[ \pi_x + \frac{1}{2}(\Delta_x + s_x) \right] \tag{31}$$
$$u_2^{\sim x} \ P_{\downarrow x}^0 \ v_1^{\sim x} = \frac{1}{\Delta_x} \left[ \overline{\beta} - \lambda_{2,x} \right] \left[ \pi_x + \frac{1}{2}(\Delta_x - s_x) \right] \tag{32}$$

*Proof of Lemma 7, part 1*:
Using (29) and (30), we have

$$\left| (u_i^{\sim x}\mathbf{1}) \ u_i^{\sim x} \ P_{\downarrow x}^0 \ v_i^{\sim x} - \pi_x\lambda_{i,x} \right|$$
$$= \ \Delta_x^{-2} \left| (1/4) \ (\Delta_x^2 - s_x^2) \ (\overline{\beta} - \lambda_{i,x}) \right.$$
$$\left. + (-1)^{i-1} \ (1/2) \ (\Delta_x + (-1)^{i-1}s_x) \ \pi_x \ [\lambda_{i,x}(1 - \lambda_{i,x}) - \overline{\beta}(\pi_x + v_x u_x)] \right|$$
$$= \ \Delta_x^{-2} \left| \overline{\beta} \ \pi_x \ [v_x u_x \ (\overline{\beta} - \lambda_{i,x}) + (1/2) \ (-1)^i \ (\Delta_x + (-1)^{i-1}s_x) \ (\pi_x + v_x u_x)] \right.$$
$$\left. + (-1)^{i-1} \ (1/2) \ (\Delta_x + (-1)^{i-1}s_x) \ \pi_x \ \lambda_{i,x}(1 - \lambda_{i,x}) \right|$$

Using $(1/2)\,(\Delta_x - s_x) = \pi_x - (1 - \lambda_{1,x})$, $(1/2)\,(\Delta_x + s_x) = 1 - \pi_x - \lambda_{2,x}$ along with $\psi_{xx} = \overline{\beta} v_x u_x$, $|\overline{\beta}| \le 1$, $1 - \pi_x \le 1$ and $|\lambda_{i,x}| \le 1$, we get

$$\left| (u_1^{\sim x} \mathbf{1})\, u_1^{\sim x}\, P_{\downarrow x}^0\, v_1^{\sim x} - \pi_x \lambda_{1,x} \right| \;\le\; \Delta_x^{-2}\,\left[ 2(2|\psi_{xx}| + \pi_x)\,\pi_x + 2\,\pi_x\,\lambda_{1,x}\,(1 - \lambda_{1,x}) \right]$$

$$\left| (u_2^{\sim x} \mathbf{1})\, u_2^{\sim x}\, P_{\downarrow x}^0\, v_2^{\sim x} - \pi_x \lambda_{2,x} \right| \;\le\; \Delta_x^{-2}\,\left[ 4(|\psi_{xx}| + \pi_x)\,\pi_x + 2\,\pi_x\,\lambda_{1,x}\,(1 - \lambda_{1,x}) \right]$$

*Proof of Lemma 7, part 2*:
Using (29) along with (31) and (32), we have

$$\left| (u_2^{\sim x} \mathbf{1})\, (u_1^{\sim x}\, P_{\downarrow x}^0\, v_2^{\sim x}) \right| = \left| \Delta_x^{-2}\, \pi_x\, \left[ \lambda_{1,x} - \overline{\beta} \right]\, \left[ \pi_x - (1 - \lambda_{1,x}) + \psi_{xx} \right] \right|$$

$$\overset{(d_1)}{\le}\; 6\,\Delta_x^{-2}\,\pi_x,$$

$$u_1^{\sim x} \mathbf{1}\, u_2^{\sim x}\, P_{\downarrow x}^0\, v_1^{\sim x} = \left| \Delta_x^{-2}\, \pi_x\, \left[ \overline{\beta} - \lambda_{2,x} \right]\, \left[ (1 - \pi_x - \lambda_{2,x}) + \psi_{xx} \right] \right|$$

$$\overset{(d_2)}{\le}\; 6\,\Delta_x^{-2}\,\pi_x,$$

where we get $(d_1)$ and $(d_2)$ using $|\overline{\beta}| \le 1$, $|\lambda_{i,x}| \le 1$, $1 - \pi_x \le 1$ and $|\psi_{xx}| \le 1$. $\qquad\square$

Using (27) and (28) in (26) to bound the absolute value of $\Gamma_x$ for $x \in A(\delta)$, we have

$$\begin{aligned}
|\Gamma_x| \;\le\;\; & 2\,(\lambda_{1,x})^{n-1}\,\Delta_x^{-2}\,\left[ 4\,(|\psi_{xx}| + \pi_x)\,\pi_x + 2\,\pi_x\,\lambda_{1,x}\,(1 - \lambda_{1,x}) \right] \\
& + (12/n)\,\Delta_x^{-3}\,\left[ (\lambda_{1,x})^n - (\lambda_{2,x})^n \right]\,\pi_x \\
\overset{(b_1)}{\le}\;\; & 2\,\Delta_x^{-2}\,\left[ 4\,(|\psi_{xx}| + \pi_x)\,\pi_x\,(\lambda_{1,x})^{n-1} + 2\,\pi_x\,(\lambda_{1,x})^n\,(1 - \lambda_{1,x}) \right] \\
& + (24/n)\,\Delta_x^{-3}\,\pi_x \\
\overset{(b_2)}{\le}\;\; & 4\,(1/n)\,\Delta_x^{-2}\,\left[ (4/c_\beta)\,(|\psi_{xx}| + \pi_x)\; + e^{-1}\,\pi_x + 6\,\Delta_x^{-1}\,\pi_x \right],
\end{aligned} \tag{33}$$

where we get $(b_1)$ by using $|\lambda_{i,x}| \le 1$, $i = 1, 2$, and $(b_2)$ by using (23) along with $\max_{p \in (0,1)} p\,(1 - cp)^n \le \min\{e^{-1}/(cn), 1/c(n+1)\}$.

We next claim that $\Delta_x$ is bounded away from 0 for $x \in A(\delta)$.
**Claim 1:** $\Delta_x \ge \beta/3$, for $x \in A(\delta)$.

*Proof.* Using $\overline{\beta} v_x u_x \ge -\pi_x$ from (16), we get

$$\beta - \pi_x + \overline{\beta} v_x u_x \ge \beta - 2\pi_x,$$
$$4\pi_x \overline{\beta} v_x u_x \ge -4\pi_x^2.$$

Using the above in the expression for $\Delta_x^2$,

$$\Delta_x^2 = (\beta - \pi_x + \overline{\beta} v_x u_x)^2 + 4\pi_x \overline{\beta} v_x u_x \;\ge\; (\beta - 2\pi_x)^2 - 4\pi_x^2 \;=\; \beta^2 - 4\beta\pi_x > \beta^2/5 > \beta^2/9, \tag{34}$$

where we use $\pi_x < \beta/5$ to get $(a)$. $\qquad\square$

Using the above lower bound on $\Delta_x$ for $x \in A(\delta)$ in (33), we get

$$|\Gamma_x| \;\le\; 36\,(1/n\beta^2)\,\left[ (4/c_\beta)\,(|\psi_{xx}| + \pi_x)\; + (e^{-1} + 18/\beta)\,\pi_x \right]. \tag{35}$$

Taking the sum of (35) over $x$ in $A(\delta)$, we get

$$\sum_{x \in A(\delta)} |\Gamma_x| \;\overset{(a_1)}{\le}\; 36\,(1/n\beta^2)\,(e^{-1} + 16/c_\beta + 34/\beta), \tag{36}$$

where we get $(a_1)$ by using $\sum_{x \in A(\delta)} \pi_x \le 1$ and Lemma 6 as $\sum_{x \in A(\delta)} |\psi_{xx}| \le \sum_{x \in \mathcal{X}} |\psi_{xx}| \le 1$.

### A.2 Other two cases

When $x \notin A(\delta)$, we require computation of the probabilities $\Pr(N_x(X^n) = 0)$ and $\Pr(N_x(X^n) = 1)$ occurring in the definition of $\Gamma_x$. $\Pr(N_x(X^n) = 0)$ can be written as

$$\Pr(N_x(X^n) = 0) \stackrel{(a)}{=} \boldsymbol{\pi}^{\sim x} (P^{\sim x})^{n-1} \mathbf{1} = (\boldsymbol{\pi}^{\sim x}) R D (S^{\sim x} RD)^{n-2} S^{\sim x} \mathbf{1}$$

$$\stackrel{(b)}{=} \begin{bmatrix} 1 - \pi_x & -(1-\beta)\pi_x v_x \end{bmatrix} (S^{\sim x} RD)^{n-2} \begin{bmatrix} 1 - \pi_x \\ -u_x \end{bmatrix}, \tag{37}$$

where we get $(a)$ by noting that the entry of the $(K \times 1)$ vector $(P^{\sim x})^{n-1} \mathbf{1}$ corresponding to any state $z \in \mathcal{X}$ is the probability of not passing through the state $x$ in the next $n - 1$ steps, given the present state is $z$, $(b)$ by using $\boldsymbol{\pi}\mathbf{1} = 1, \boldsymbol{\pi} v = 0$, and $u\mathbf{1} = 0$.

$\Pr(N_x(X^n) = 1)$ can be written as

$$\Pr(N_x(X^n) = 1) = \sum_{m=1}^{n} \Pr\big(X_m = x, N_x(X_1^{m-1}) = N_x(X_{m+1}^n) = 0\big)$$

$$\stackrel{(a)}{=} \sum_{m=1}^{n} \Pr(X_m = x, N_x(X_1^{m-1}) = 0) \Pr(N_x(X_{m+1}^n) = 0 | X_m = x)$$

$$\stackrel{(b)}{=} \pi_x^{-1} \sum_{m=1}^{n} \Pr(X_1 = x, N_x(X_2^m) = 0) \Pr(X_1 = x, N_x(X_2^{n-m+1}) = 0), \tag{38}$$

where we get $(a)$ by using the Markov property, $(b)$ using

$$\Pr(X_m{=}x, N_x(X_1^{m-1}) = 0) = \Pr(X_1{=}x, N_x(X_2^m) = 0) \tag{39}$$

and noting that $\Pr(N_x(X_{m+1}^n) = 0 | X_m = x) = \pi_x^{-1} \Pr(X_1 = x, N_x(X_2^{n-m+1}) = 0)$. Now,

$$\Pr(X_1 = x, N_x(X_2^m) = 0) = \pi_x e_x (P^{\sim x})^{m-1} \mathbf{1} = \pi_x e_x R D (S^{\sim x} RD)^{m-2} S^{\sim x} \mathbf{1}$$

$$= \pi_x \begin{bmatrix} 1 & \overline{\beta} v_x \end{bmatrix} (S^{\sim x} RD)^{m-2} \begin{bmatrix} 1 - \pi_x \\ -u_x \end{bmatrix}, \tag{40}$$

where $e_x$ is a $1 \times K$ vector with $x$-th entry as 1 and all other entries as 0.

The expressions for $\Pr(N_x(X^n) = 0)$ and $\Pr(N_x(X^n) = 1)$ involve powers of the $2 \times 2$ matrix $S^{\sim x} RD$. Using the eigen decomposition, we find expressions for terms in powers of $S^{\sim x} RD$ and derive bounds for $\Pr(N_x(X^n) = 0)$, $\Pr(N_x(X^n) = 1)$ and $|\Gamma_x|$. This is done differently for the two remaining cases.

### A.2.1 Case 2: iid-like letters

Since $P_{xx} = \pi_x + (1 - \beta)v_x u_x$, $\pi_x = P_{xx}$ implies that $u_x = 0$ or $v_x = 0$. For this scenario, the powers of $S^{\sim x} RD$ simplify as shown in the following lemma.

**Lemma 8.**     *1. For $x \in \mathcal{X}$, with $v_x = u_x = 0, (S^{\sim x} RD)^l = \begin{bmatrix} \overline{\pi_x}^l & 0 \\ 0 & \overline{\beta}^l \end{bmatrix}$.*

    *2. For $x \in \mathcal{X}$, with $\pi_x = P_{xx} = \beta$,*

        *(a) $v_x = 0, u_x \neq 0$:*

$$(S^{\sim x} RD)^l = \begin{bmatrix} \overline{\pi_x}^l & 0 \\ -l u_x \overline{\pi_x}^{l-1} & \overline{\pi_x}^l \end{bmatrix}$$

        *(b) $v_x \neq 0, u_x = 0$:*

$$(S^{\sim x} RD)^l = \begin{bmatrix} \overline{\pi_x}^l & -l \pi_x v_x \overline{\pi_x}^l \\ 0 & \overline{\pi_x}^l \end{bmatrix}$$

*3. For $x \in \mathcal{X}$, with $\pi_x = P_{xx} \neq \beta$,*

    *(a) $v_x = 0, u_x \neq 0$:*

$$(S^{\sim x}RD)^l = \begin{bmatrix} 1 & 0 \\ \frac{u_x}{\pi_x - \beta} & 1 \end{bmatrix} \begin{bmatrix} \overline{\pi_x}^l & 0 \\ 0 & \overline{\beta}^l \end{bmatrix} \begin{bmatrix} 1 & 0 \\ -\frac{u_x}{\pi_x - \beta} & 1 \end{bmatrix} \tag{41}$$

    *(b) $v_x \neq 0, u_x = 0$:*

$$(S^{\sim x}RD)^l = \begin{bmatrix} 1 & -\frac{\overline{\beta}\pi_x v_x}{\pi_x - \beta} \\ 0 & 1 \end{bmatrix} \begin{bmatrix} \overline{\pi_x}^l & 0 \\ 0 & \overline{\beta}^l \end{bmatrix} \begin{bmatrix} 1 & \frac{\overline{\beta}\pi_x v_x}{\pi_x - \beta} \\ 0 & 1 \end{bmatrix} \tag{42}$$

*Proof.* Since $SR = I$, we have $S^{\sim x}R = \begin{bmatrix} \overline{\pi_x} & -\pi_x v_x \\ -u_x & 1 - v_x u_x \end{bmatrix}$ and

$$S^{\sim x}RD = \begin{bmatrix} \overline{\pi_x} & -\overline{\beta}\pi_x v_x \\ -u_x & \overline{\beta}(1 - v_x u_x) \end{bmatrix} \tag{43}$$

1. We substitute $v_x = u_x = 0$ in (43) and raise the power on both sides to $l$.

2. (a) Substituting $v_x = 0$, $\beta = \pi_x$ in (43), we get $S^{\sim x}RD = \begin{bmatrix} \overline{\pi_x} & 0 \\ -u_x & \overline{\pi_x} \end{bmatrix}$. Note that this matrix is not diagonalizable since both its eigenvalues are equal to $\overline{\pi_x}$ with $[0\ 1]^T$ as the only non-trivial eigenvector. Using induction on the exponent $l$, we get $(S^{\sim x}RD)^l = \begin{bmatrix} \overline{\pi_x}^l & 0 \\ -l u_x \overline{\pi_x}^{l-1} & \overline{\pi_x}^l \end{bmatrix}$.

   (b) Substituting $u_x = 0$, $\beta = \pi_x$ in (43), we get $S^{\sim x}RD = \begin{bmatrix} \overline{\pi_x} & -\overline{\pi_x}\pi_x v_x \\ 0 & \overline{\pi_x} \end{bmatrix}$. Similar to the above case, this matrix is also not diagonalizable with both its eigenvalues as $\overline{\pi_x}$ and $[1\ 0]^T$ as the only non-trivial eigenvector. Using induction on the exponent $l$, we get $(S^{\sim x}RD)^l = \begin{bmatrix} \overline{\pi_x}^l & -l\pi_x v_x \overline{\pi_x}^l \\ 0 & \overline{\pi_x}^l \end{bmatrix}$.

3. (a) Substituting $v_x = 0$ in (43), we get $S^{\sim x}RD = \begin{bmatrix} \overline{\pi_x} & 0 \\ -u_x & \overline{\beta} \end{bmatrix}$. We observe that $\overline{\pi_x}, \overline{\beta}$ are the eigenvalues of $S^{\sim x}RD$ with $\begin{bmatrix} 1 \\ \frac{u_x}{\pi_x - \beta} \end{bmatrix}$ and $\begin{bmatrix} 0 \\ 1 \end{bmatrix}$ as their respective right eigenvectors resulting in the diagonalised form in (41).

   (b) Substituting $u_x = 0$ in (43), we get $S^{\sim x}RD = \begin{bmatrix} \overline{\pi_x} & -\overline{\beta}\pi_x v_x \\ 0 & \overline{\beta} \end{bmatrix}$. We observe that $\overline{\pi_x}, \overline{\beta}$ are the eigenvalues of $S^{\sim x}RD$ with $\begin{bmatrix} 1 \\ 0 \end{bmatrix}$ and $\begin{bmatrix} -\frac{\overline{\beta}\pi_x v_x}{\pi_x - \beta} \\ 1 \end{bmatrix}$ as their respective right eigenvectors resulting in the diagonalised form in (42).

$\square$

Substituting the corresponding form of $(S^{\sim x}RD)^{n-2}$ in (37) and (40) gives

$$\Pr(N_x(X^n) = 0) = (1 - \pi_x)^n, \tag{44}$$
$$\Pr(X_1 = x, N_x(X_2^m) = 0) = \pi_x (1 - \pi_x)^{m-1}.$$

Using the above in (38), we get

$$\Pr(N_x(X^n) = 1) = n\pi_x (1 - \pi_x)^{n-1}.$$

So, we have (in a manner reminiscent of the iid case)

$$|\Gamma_x| = \pi_x(1-\pi_x)^{n-1} - \pi_x(1-\pi_x)^n = \pi_x^2(1-\pi_x)^{n-1}] \le \pi_x \max_p p(1-p)^{n-1} = \pi_x \, O(1/n).$$

This results in the following bound:

$$\sum_{x \in A_0} |\Gamma_x| = O(1/n). \tag{45}$$

### A.2.2 Case 3: Frequent letters

In this case, $\pi_x > \delta$ and the letters $x$ are likely to occur multiple times. So, $\Pr(N_x(X^n) = 0)$ and $\Pr(N_x(X^n) = 1)$ will both be small, and we bound them both. Further, since $P_{xx} \ne \pi_x$, we have that $u_x \ne 0$ and $v_x \ne 0$. For this scenario, the powers of $S^{\sim x}RD$ simplify as shown in the following lemma.

**Lemma 9.** *For $x \in \mathcal{X}$, with $v_x \ne 0, u_x \ne 0$,*

$$(S^{\sim x}RD)^l = V^{\sim x} \begin{bmatrix} (\lambda_{1,x})^l & 0 \\ 0 & (\lambda_{2,x})^l \end{bmatrix} (V^{\sim x})^{-1} \tag{46}$$

$$where \ V^{\sim x} = \begin{bmatrix} 1 & 1 \\ -\dfrac{\Delta_x - s_x}{2\overline{\beta}\pi_x v_x} & \dfrac{\Delta_x + s_x}{2\overline{\beta}\pi_x v_x} \end{bmatrix},$$

$s_x = 1 - \pi_x - \overline{\beta}(1 - v_x u_x).$

*Proof.* Solving $\det(S^{\sim x}RD - \lambda I) = 0$, we get $\lambda_{i,x} = 0.5\big(\overline{\pi}_x + \overline{\beta}(1 - v_x u_x) + (-1)^{i+1}\Delta_x\big)$, $i = 1, 2$, as the eigenvalues of $S^{\sim x}RD$ with $\begin{bmatrix} 1 \\ -(\Delta_x - s_x)/(2\overline{\beta}\pi_x v_x) \end{bmatrix}, \begin{bmatrix} 1 \\ (\Delta_x + s_x)/(2\overline{\beta}\pi_x v_x) \end{bmatrix}$ as their respective right eigenvectors, where $\Delta_x^2 = s_x^2 + 4\overline{\beta}\pi_x v_x u_x, s_x = 1 - \pi_x - \overline{\beta}(1 - v_x u_x)$, resulting in the diagonalised form in (46). $\qquad \square$

Substituting the diagonalized form of $(S^{\sim x}RD)^{n-2}$ from (46) into (37), (40) and simplifying, we get

$$\Pr(N_x(X^n) = 0) = \frac{1}{2}\bigg[ (\lambda_{1,x})^n + (\lambda_{2,x})^n \bigg] + \frac{s_x}{2}\bigg[ \sum_{l=0}^{n-1} (\lambda_{1,x})^{n-1-l}(\lambda_{2,x})^l \bigg], \tag{47}$$

$$\Pr(X_1 = x, N_x(X_2^m) = 0) = \frac{1}{2}\pi_x\bigg[ (\lambda_{1,x})^{m-1} + (\lambda_{2,x})^{m-1} \bigg] + \pi_x\bigg[ \frac{s_x}{2} - \overline{\beta}v_x u_x \bigg]\bigg[ \sum_{l=0}^{m-2} (\lambda_{1,x})^{m-2-l}(\lambda_{2,x})^l \bigg]. \tag{48}$$

**Claim 2:** $|s_x| \le 3$

*Proof.* Using $|\overline{\beta}| \le 1$, $\pi_x \le 1$, and $\overline{\beta}v_x u_x \le 1 - \pi_x$ from (16) in $s_x = \overline{\pi}_x - \overline{\beta} + \overline{\beta}v_x u_x$, we get $|s_x| \le 3$. $\qquad \square$

Using triangle inequality on the R.H.S of (47), we get

$$\begin{aligned}
\Pr(N_x(X^n) = 0) &\le \frac{1}{2}\bigg[ (|\lambda_{1,x}|)^n + (|\lambda_{2,x}|)^n \bigg] + \frac{|s_x|}{2}\bigg[ \sum_{l=0}^{n-1} (|\lambda_{1,x}|)^{n-1-l} \, (|\lambda_{2,x}|)^l \bigg] \\
&\overset{(a)}{\le} \left( e^{-0.5c_\beta\pi_x} + \frac{n|s_x|}{2} \right) e^{-0.5(n-1)c_\beta\pi_x} \\
&\overset{(b)}{\le} \left( 1 + \frac{3n}{2} \right) e^{-0.5(n-1)c_\beta\pi_x} \overset{(c)}{\le} \left( 1 + \frac{3n}{2} \right) e^{-0.5(n-1)c_\beta\delta}, \tag{49}
\end{aligned}$$

where we use (23) along with $1 - z \le e^{-z}$ to get $(a)$, $e^{-z} \le 1$ and the above Claim 2 to get $(b)$, and $\pi_x \ge \delta$ for $x \in \overline{A}(\delta)$ to get $(c)$.

**Claim 3:** For $x$ in $\overline{A}(\delta)$, $m = 1, \ldots, n$,

$$\Pr(X_1 = x, N_x(X_2^m) = 0) \leq \pi_x \ (1 + (5/2) \ (m - 1)) \ \exp\{-(m - 2) \ c_\beta \ \delta/2\}.$$

*Proof.* To get the above bound, we bound (48) using $|s_x| \leq 3$ (Claim 2 above), $|\overline{\beta} v_x u_x| \leq 1$, and $|\lambda_{i,x}| \leq 1 - c_\beta \delta/2$ (from (23) along with $1 - t \leq e^{-t}$). $\qquad \square$

Using Claim 3 in (38), we get

$$\Pr(N_x(X^n) = 1) \leq \sum_{m=1}^{n} \pi_x \ (1 + (5/2) \ (m - 1)) \ \exp\{-(m - 2) \ c_\beta \ \delta/2\}$$

$$(1 + (5/2) \ (n - m)) \ \exp\{-(n - m - 1) \ c_\beta \ \delta/2\}$$

$$= \ n \ \pi_x \ g(n) \ \exp\{-(n - 3) \ c_\beta \ \delta/2\},$$

where $g(n) = [1 + (5/2) \ (n - 1) \ (1 + (5/12) \ (n - 2))]$. So,

$$|\Gamma_x| = \left|\frac{1}{n}\Pr(N_x(X^n) = 1) - \pi_x\Pr(N_x(X^n) = 0)\right| \leq \frac{1}{n}\Pr(N_x(X^n) = 1) + \pi_x\Pr(N_x(X^n) = 0)$$

$$\leq \ \pi_x \ \Big[g(n) \ \exp\{-(n - 3) \ c_\beta \ \delta/2\} + (1 + (3/2) \ n) \ \exp\{-(n - 1) \ c_\beta \ \delta/2\}\Big].$$

Summing over $x$ in $\overline{A}(\delta)$ and using $\sum_{x \in \overline{A}(\delta)} \pi_x \leq 1$, we get

$$\sum_{x \in \overline{A}(\delta)} |\Gamma_x| \leq \Big[g(n) \ \exp\{-(n - 3) \ c_\beta \ \delta/2\} + (1 + (3/2) \ n) \ \exp\{-(n - 1) \ c_\beta \ \delta/2\}\Big]. \qquad (50)$$

### A.3 Combining all cases

Taking the sum of (36), (45) and (50), and choosing $\delta = (6/c_\beta) \ (\ln n)/(n - 3)$, we get

$$\left|E[\widehat{M}_0^{\mathrm{GT}}(X^n) - M_0(\boldsymbol{\pi}, X^n)]\right| \leq 36 \ (1/n\beta^2) \ (1 + 16/c_\beta + 18/\beta) + O(1/n),$$

for $\beta \geq \Big[30(\ln n)/(n - 3)\Big]^{1/2}$ (since $\delta \leq \beta/5$). This concludes the proof of Theorem 3.

### A.4 Proofs of some lemmas

#### A.4.1 Proof of Lemma 6

1. We first consider the case with $a \geq 1$.

$$\sum_{x \in \mathcal{X}} (\pi_x)^a \ |\psi_{xx}^b| \ |\psi_{yx}^c| \ \overset{(f)}{\leq} \ \sum_{x \in \mathcal{X}} (\pi_x)^a \ \overset{(g)}{\leq} \ \sum_{x \in \mathcal{X}} \pi_x \ = \ 1,$$

   where we get $(f)$ by using $|\psi_{xx}|, |\psi_{yx}| \leq 1$ (from (16)) and $(g)$ by using $\pi_x^a \leq \pi_x$.

2. We now consider the case with $a = 0$. Since $a + b + c \geq 1$, atleast one of $b, c$ must be $\geq 1$.

   (a) Say $c \geq 1$.

   $$\sum_{x \in \mathcal{X}} |\psi_{xx}^b| \ |\psi_{yx}^c| \ \overset{(a)}{\leq} \ \sum_{x \in \mathcal{X}} |\psi_{yx}| \ = \ \sum_{x \in \mathcal{X}:\psi_{yx}<0} |\psi_{yx}| + \sum_{x \in \mathcal{X}:\psi_{yx}\geq 0} \psi_{yx}$$

   $$\overset{(b)}{=} \ 2 \sum_{x \in \mathcal{X}:\psi_{yx}<0} |\psi_{yx}| \ \overset{(c)}{\leq} \ 2 \sum_{x \in \mathcal{X}:\psi_{yx}<0} \pi_x \ \leq \ 2,$$

   where we get $(a)$ by using $|\psi_{xx}|, |\psi_{yx}| \leq 1$ (from (16)), $(b)$ by using $\sum_{x \in \mathcal{X}} \psi_{yx} = 0$, and $(c)$ by using $-\psi_{yx} \leq \pi_x$ (from (16)).

(b) Say $b \geq 1$.

$$\sum_{x \in \mathcal{X}} |\psi_{xx}^b| \, |\psi_{yx}^c| \overset{(a)}{\leq} \sum_{x \in \mathcal{X}} |\psi_{xx}| = \sum_{x \in \mathcal{X}: \psi_{xx} < 0} |\psi_{xx}| + \sum_{x \in \mathcal{X}: \psi_{xx} \geq 0} \psi_{xx}$$

$$\overset{(b)}{=} (1 - \beta) + 2 \sum_{x \in \mathcal{X}: \psi_{xx} < 0} |\psi_{xx}|$$

$$\overset{(c)}{\leq} 1 + 2 \sum_{x \in \mathcal{X}: \psi_{xx} < 0} \pi_x \leq 3,$$

where we get $(a)$ by using $|\psi_{xx}|, |\psi_{yx}| \leq 1$ (from (16)), $(b)$ by using $\sum_{x \in \mathcal{X}} \psi_{xx} = (1 - \beta)$, and $(c)$ by using $-\psi_{xx} \leq \pi_x$ (from (16)).

This completes the proof.

### A.4.2 Proof of Lemma 5

To prove (21), we first start with the expression for $\Delta_x^2$ and bound it in the following way:

$$\Delta_x^2 = [(1 - \pi_x) - (1 - \beta)(1 - v_x u_x)]^2 + 4(1 - \beta)\pi_x v_x u_x$$

$$= [(1 - \pi_x) + (1 - \beta)(1 - v_x u_x)]^2 - 4(1 - \beta)(1 - \pi_x - v_x u_x)$$

$$\overset{(a)}{=} [\beta + (1 - \beta)\pi_x + (1 - \beta)v_x u_x]^2 + \pi_x^2(\beta^2 + 2\beta(1 - \beta)) + 2\beta\pi_x[(\beta - 2) + (1 - \beta)v_x u_x]$$

$$\overset{(b)}{\leq} [\beta(1 - \pi_x) + P_{xx}]^2 + \pi_x^2(\beta^2 + 2\beta(1 - \beta)) + 2\beta\pi_x(\beta - 2) + 2\beta\pi_x(1 - \pi_x)$$

$$= [\beta(1 - \pi_x) + P_{xx}]^2 - \beta^2 \pi_x^2 + 2\beta\pi_x(\beta - 1)$$

$$\overset{(c)}{\leq} [\beta(1 - \pi_x) + P_{xx}]^2$$

where $(a)$ follows by using $[(1 - \pi_x) + (1 - \beta)(1 - v_x u_x)]^2 = [\beta + (1 - \beta)\pi_x + (1 - \beta)v_x u_x + \beta\pi_x - 2]^2$ and simplyfying, $(b)$ follows by using $P_{xx} = \pi_x + (1 - \beta)v_x u_x, (1 - \beta)v_x u_x \leq 1 - \pi_x$ from (16) and $(c)$ follows from $\beta \leq 1$. Since $\beta(1 - \pi_x) + P_{xx} \geq 0$, we get $\Delta_x \in [-(\beta(1 - \pi_x) + P_{xx}), (\beta(1 - \pi_x) + P_{xx})]$. This completes the proof of (21).

To prove (22), we again start with the expression for $\Delta_x^2$ but bound it in a different way as shown below:

$$\Delta_x^2 = [(1 - \pi_x) - (1 - \beta)(1 - v_x u_x)]^2 + 4(1 - \beta)\pi_x v_x u_x$$

$$= [\beta + (1 - \beta)v_x u_x - \pi_x]^2 + 4(1 - \beta)\pi_x v_x u_x$$

$$= [\beta + (1 - \beta)v_x u_x]^2 - 2\beta\pi_x + \pi_x^2 + 2(1 - \beta)\pi_x v_x u_x$$

$$\overset{(a)}{\leq} [\beta + (1 - \beta)v_x u_x]^2 - 2\beta\pi_x + \pi_x^2 + 2\pi_x(1 - \pi_x)$$

$$= [\beta + (1 - \beta)v_x u_x]^2 + 2\pi_x(1 - \beta) - \pi_x^2$$

$$\overset{(b)}{\leq} [\beta + (1 - \beta)v_x u_x]^2$$

where $(a)$ follows by using $(1 - \beta)v_x u_x \leq 1 - \pi_x$ from (16) and $(b)$ follows from $\beta \geq 1$. Since $\beta + (1 - \beta)v_x u_x$ is positive for $\beta \geq 1$, we get $\Delta_x \in [-(\beta + (1 - \beta)v_x u_x), (\beta + (1 - \beta)v_x u_x)]$. This completes the proof of (22).

Using (21) and (22) in the expression $\lambda_{i,x} = \frac{1}{2}\left((1 - \pi_x) + (1 - \beta)(1 - v_x u_x) + (-1)^{i+1}\Delta_x\right), \; i = 1, 2$, we get $\lambda_{i,x} \leq 1 - 0.5\beta\pi_x, \beta \in [0, 1]$ and $\lambda_{i,x} \leq 1 - 0.5\pi_x, \beta \in [1, 2]$ for $i = 1, 2$. Using $\frac{\beta^2}{\beta+2} \leq \beta$ for $\beta \in [0, 1]$ and $\beta^2 \leq \beta + 2$ for $\beta \in [1, 2]$ completes the proof of Lemma 5.

## B    Proof of Theorem 4, Upper bound

In this appendix, we provide proof for the upper bound in theorem 4.

The minimax risk $R_n^*(P_{2,\beta})$ is upper bounded by the worst-case risk (over $\mathcal{P}_{2,\beta}$) of the Good-Turing estimator. Our next lemma gives an expression for the MSE of the GT estimator.

**Lemma 10.** *Consider a stationary Markov chain $X^n$ with state distribution $\boldsymbol{\pi}$. Let $Q_x^n(a) \triangleq Pr(N_x(X^n) = a)$, and $Q_{x,y}^n(a,b) \triangleq Pr(N_x(X^n) = a, N_y(X^n) = b)$ for $x, y \in \mathcal{X}$ and $a, b \in \{0, 1, \ldots\}$.*

$$
E[(M_0(\boldsymbol{\pi}, X^n) - \widehat{M}_0^{GT}(X^n))^2]
$$
$$
= \sum_{x \in \mathcal{X}} \left( \pi_x^2 \, Q_x^n(0) + (1/n)^2 \, Q_x^n(1) \right) + \sum_{x \in \mathcal{X}} \sum_{y \in \mathcal{X}, y \neq x} T_{xy}^n - \frac{1}{n^2(n-1)} \, Q_{x,y}^n(1,1) \tag{51}
$$

*where $T_{xy}^n \triangleq \pi_x \pi_y \, Q_{x,y}^n(0,0) - \frac{1}{n}(\pi_y \, Q_{x,y}^n(1,0) + \pi_x \, Q_{x,y}^n(0,1)) + \frac{1}{n(n-1)} \, Q_{x,y}^n(1,1).$*

*Proof.* Substituting $M_0(\boldsymbol{\pi}, X^n) = \sum_{x \in \mathcal{X}} \pi_x I(N_x = 0)$ and $\widehat{M}_0^{\mathrm{GT}}(X^n) = (1/n) \sum_{x \in \mathcal{X}} I(N_x = 1)$ into $E[(M_0(X^n, \boldsymbol{\pi}) - \widehat{M}_0^{\mathrm{GT}}(X^n))^2]$ and taking expectation of each term in the square of the summation,

$$
E[(M_0 - \widehat{M}_0^{\mathrm{GT}})^2] = E\left[ \left( \sum_{x \in \mathcal{X}} \pi_x I(N_x = 0) - (1/n)I(N_x = 1) \right)^2 \right]
$$
$$
= \sum_{x \in \mathcal{X}} \left( \pi_x^2 \, \Pr(N_x = 0) + (1/n)^2 \, \Pr(N_x = 1) \right)
$$
$$
+ \sum_{x \in \mathcal{X}} \sum_{y \in \mathcal{X}, y \neq x} \pi_x \pi_y \Pr(N_x = N_y = 0) - \frac{1}{n} \pi_y \Pr(N_x = 1, N_y = 0)
$$
$$
- \frac{1}{n} \pi_x \Pr(N_x = 0, N_y = 1) + \frac{1}{n^2} \Pr(N_x = N_y = 1)
$$
$$
\overset{(a)}{=} \sum_{x \in \mathcal{X}} \left( \pi_x^2 \, Q_x^n(0) + (1/n)^2 \, Q_x^n(1) \right) - \sum_{x \in \mathcal{X}} \sum_{y \in \mathcal{X}, y \neq x} \frac{1}{n^2(n-1)} \, Q_{x,y}^n(1,1)
$$
$$
+ \sum_{x \in \mathcal{X}} \sum_{y \in \mathcal{X}, y \neq x} \pi_x \pi_y \, Q_{x,y}^n(0,0) - \frac{1}{n} \left( \pi_y \, Q_{x,y}^n(1,0) + \pi_x \, Q_{x,y}^n(0,1) \right)
$$
$$
+ \frac{1}{n(n-1)} \, Q_{x,y}^n(1,1), \tag{52}
$$

where we get $(a)$ by expanding $1/n^2$ as $1/n^2 = \frac{1}{n(n-1)} - \frac{1}{n^2(n-1)}$ and using the definitions of $Q_x^n(a)$, $Q_{x,y}^n(a,b)$. Using the definition of $T_{x,y}^n$ in (52) completes the proof of (51). $\qquad\square$

To bound $E[(M_0 - \widehat{M}_0^{\mathrm{GT}})^2]$, we begin with the following expression for $Q_x^n(1)$.

$$
Q_x^n(1) = \Pr(N_x(X^n) = 1) = \sum_{l=1}^{n} \Pr(X_l = x; X_m \neq x, m \neq l) \leq \sum_{l=1}^{n} \Pr(X_l = x) = n\pi_x. \tag{53}
$$

This implies

$$
(1/n^2) \sum_{x \in \mathcal{X}} Q_x^n(1) \leq 1/n \tag{54}
$$

Similarly,

$$
Q_{x,y}^n(1,1) = \Pr(N_x = N_y = 1) = \sum_{l_1=1}^{n-1} \sum_{l_2=l_1+1}^{n} \Pr(X_{l_1} = x, X_{l_2} = y, X_m \neq x, y \; ; m \neq l_1, l_2)
$$
$$
+ \sum_{l_1=1}^{n-1} \sum_{l_2=l_1+1}^{n} \Pr(X_{l_1} = y, X_{l_2} = x, X_m \neq x, y \; ; m \neq l_1, l_2) \tag{55}
$$

For $l_2 > l_1 \geq 1$,

$$\Pr(X_{l_1} = x, X_{l_2} = y, X_m \neq x, y \; ; m \neq l_1, l_2) \; \leq \; \Pr(X_{l_1} = x, X_{l_2} = y)$$
$$= \; \pi_x \, \Pr(X_{l_2} = y | X_{l_1} = x) \tag{56}$$

$$\text{Similarly, } \Pr(X_{l_1} = y, X_{l_2} = x, X_m \neq x, y \; ; m \neq l_1, l_2) \; \leq \; \pi_y \, \Pr(X_{l_2} = x | X_{l_1} = y) \tag{57}$$

Plugging (56) and (57) into (55),

$$Q_{x,y}^n(1,1) \; \leq \; \sum_{l_1=1}^{n-1} \sum_{l_2=l_1+1}^{n} \pi_x \, \Pr(X_{l_2} = y | X_{l_1} = x)$$
$$+ \sum_{l_1=1}^{n-1} \sum_{l_2=l_1+1}^{n} \pi_y \, \Pr(X_{l_2} = x | X_{l_1} = y)$$

Since $\sum_{x \in \mathcal{X}} \sum_{y \in \mathcal{X}, y \neq x} \pi_x \, \Pr(X_{l_2} = y | X_{l_1} = x) \; \leq \; \sum_{x \in \mathcal{X}} \pi_x \; \leq \; 1$, we get $\sum_{x \in \mathcal{X}} \sum_{y \in \mathcal{X}, y \neq x} Q_{x,y}^n(1,1) \; \leq \; n(n-1)$. Therefore,

$$\frac{1}{n^2(n-1)} \sum_{x \in \mathcal{X}} \sum_{y \in \mathcal{X}, y \neq x} Q_{x,y}^n(1,1) \; \leq \; 1/n \tag{58}$$

To bound $\sum_{x \in \mathcal{X}} \pi_x^2 \, Q_x^n(0)$, we consider the sets $A_0, A(\delta)$ and $\overline{A}(\delta)$ (with $0 \leq \delta < \beta/5$) defined in section A, that make up $\mathcal{X}$.

For $x \in A(\delta)$, we have

$$Q_x^n(0) = \Pr(N_x(X^n) = 0) = (1/2)\left[ (\lambda_{1,x})^n + (\lambda_{2,x})^n \right] + (s_x/2)\left[ \sum_{l=0}^{n-1} (\lambda_{1,x})^{n-1-l} (\lambda_{2,x})^l \right]$$

from (47).

Since the claim 1 (in section A.1) implies $\lambda_{1,x} - \lambda_{2,x} = \Delta_x > 0$ for $x \in A(\delta)$, $\beta \in (0,2]$, we get

$$\sum_{l=0}^{n-1} (\lambda_{1,x})^{n-1-l} (\lambda_{2,x})^l \; = \; \frac{1}{\Delta_x}\left[ (\lambda_{1,x})^n - (\lambda_{2,x})^n \right], x \in A(\delta) \tag{59}$$

Substituting (59) in (47) and applying triangle inequality on the absolute value of the R.H.S, we get

$$Q_x^n(0) \; \leq \; \frac{1}{2}\left( 1 + \frac{|s_x|}{\Delta_x} \right)\left[ (|\lambda_{1,x}|)^n + (|\lambda_{2,x}|)^n \right]$$
$$\overset{(a)}{\leq} \; \left( 1 + \frac{|s_x|}{\Delta_x} \right)\left( 1 - \frac{c_\beta}{2}\pi_x \right)^n \overset{(b)}{\leq} \left( 1 + \frac{9}{\beta} \right)\left( 1 - \frac{c_\beta}{2}\pi_x \right)^n \tag{60}$$

where we use (23) to get $(a)$ and the claims 1 and 2 from section A to get $(b)$. Therefore, for $x \in A(\delta)$,

$$\sum_{x \in A(\delta)} \pi_x^2 \, Q_x^n(0) \; \overset{(a_1)}{\leq} \; \left( 1 + \frac{9}{\beta} \right) \sum_{x \in A(\delta)} \pi_x^2 \left( 1 - \frac{c_\beta}{2}\pi_x \right)^n$$
$$\overset{(b_1)}{\leq} \; 2\left( 1 + \frac{9}{\beta} \right)(1/nc_\beta) \sum_{x \in A(\delta)} \pi_x \overset{(c_1)}{\leq} 2\left( 1 + \frac{9}{\beta} \right)(1/nc_\beta) \tag{61}$$

where we get $(a_1)$ by using (60), $(b_1)$ by using $\max_{p \in (0,1)} p\,(1 - cp)^n = \frac{1}{c(n+1)}\left( 1 - \frac{1}{n+1} \right)^n \leq \frac{1}{cn}$, and $(c_1)$ by using $\sum_{x \in A(\delta)} \pi_x \leq 1$.

For $x \in \overline{A}(\delta)$,

$$\sum_{x \in \overline{A}(\delta)} \pi_x^2 \, Q_x^n(0) \overset{(a_2)}{\leq} \left(1 + \frac{3n}{2}\right) \, e^{-0.5(n-1)c_\beta \delta} \sum_{x \in \overline{A}(\delta)} \pi_x^2 \overset{(c_2)}{\leq} \left(1 + \frac{3n}{2}\right) \, e^{-0.5(n-1)c_\beta \delta} \tag{62}$$

where we get $(a_2)$ by using (49), and $(c_2)$ by using $\sum_{x \in \overline{A}(\delta)} \pi_x^2 \leq 1$.

For $x \in A_0(\delta)$,

$$\sum_{x \in A_0} \pi_x^2 \, Q_x^n(0) \overset{(a_0)}{=} \sum_{x \in A_0} \pi_x^2 \, (1 - \pi_x)^n \overset{(b_0)}{\leq} 1/n \sum_{x \in A_0} \pi_x \overset{(c_0)}{\leq} 1/n \tag{63}$$

where we get $(a_0)$ by using (44), $(b_0)$ by using $\max_{p \in (0,1)} p \, (1 - p)^n = \frac{1}{(n+1)} \left(1 - \frac{1}{n+1}\right)^n \leq \frac{1}{n}$, and $(c_0)$ by using $\sum_{x \in A_0} \pi_x \leq 1$.

Since $\mathcal{X} = A(\delta) \cup \overline{A}(\delta) \cup A_0$,

$$\sum_{x \in \mathcal{X}} \pi_x^2 \, Q_x^n(0) \;=\; \sum_{x \in A_0} \pi_x^2 \, Q_x^n(0) + \sum_{x \in A(\delta)} \pi_x^2 \, Q_x^n(0) + \sum_{x \in \overline{A}(\delta)} \pi_x^2 \, Q_x^n(0)$$

$$\overset{(a)}{\leq} \; 1/n + 2 \left(1 + \frac{9}{\beta}\right) \, (1/nc_\beta) + \left(1 + \frac{3n}{2}\right) \, e^{-0.5(n-1)c_\beta \delta}$$

where we get $(a)$ by combining (61), (62) and (63).

Choosing $\delta = (8/c_\beta) \, (\ln n)/(n-1)$ and using $n - 5 < n - 1$, we get

$$\sum_{x \in \mathcal{X}} \pi_x^2 \, Q_x^n(0) \;\leq\; 2 \, \left(1 + \frac{9}{\beta}\right) \, (1/nc_\beta) + \, O(1/n), \tag{64}$$

where $c_\beta = \beta$ for $\beta \in (0,1]$ and $c_\beta = 1$ for $\beta \in [1,2]$.

To bound $\sum_{x \in \mathcal{X}} \sum_{y \in \mathcal{X}, y \neq x} T_{xy}^n$, we divide $\{(x,y) : x \in \mathcal{X}, y \in \mathcal{X}, x \neq y\}$ into two sets:

$$B(\delta) \;\triangleq\; \{(x,y) : x, y \in \mathcal{X}, \; x \neq y, \; \pi_x + \pi_y < \delta\}$$
$$\overline{B}(\delta) \;\triangleq\; \{(x,y) : x, y \in \mathcal{X}, \; x \neq y, \; \pi_x + \pi_y \geq \delta\}$$

with $0 \leq \delta < \beta/5$. Similar to the sets $A(\delta)$ and $\overline{A}(\delta)$, defined in section A, the ordered pairs in $\overline{B}(\delta)$ have atleast one frequent letter that is likely to be seen in $X^n$, while the set $B(\delta)$ contains ordered pairs of rare letters that are less likely to be seen in $X^n$ and hence contribute more to the missing mass $M_0(\boldsymbol{\pi}, X^n)$.

In the following lemma, we provide bounds on $T_{xy}^n$ for $(x,y)$ in $B(\delta)$, $\overline{B}(\delta)$.

**Lemma 11.** *For a rank-2 diagonalizable t.p.m $P$ with spectral gap $\beta$, and $\delta \in [0, \beta/5)$,*

    *1. For $(x,y)$ in $B(\delta)$,*

$$T_{xy}^n \;\leq\; q_{xy}' \, (1/\beta^3) \, (\pi_x + \pi_y) \, \left(1 - \frac{\beta^2}{2(\beta+2)}(\pi_x + \pi_y)\right)^{n-2}$$
$$+ \, q_{xy} \, (1/n\beta^4) \, \left[1 + \frac{6}{\beta(n-1)}\right] \tag{65}$$

    *2. For $(x,y)$ in $\overline{B}(\delta)$,*

$$T_{xy}^n \;\leq\; q_{xy}'' \, O(n^3) \, \exp\left\{-(n-5) \, \frac{\beta^2}{2(\beta+2)} \, \delta\right\} \tag{66}$$

where the positive constants $q_{xy}, q'_{xy},$ and $q''_{xy}$ are such that $\sum_{(x,y)\in\mathcal{X}^2} q_{xy}, \sum_{(x,y)\in\mathcal{X}^2} q'_{xy}$ and $\sum_{(x,y)\in\mathcal{X}^2} q''_{xy}$ are constant w.r.t. $n$.

*Proof.* We get (65) by using a method similar to the upper bound on the letter wise bias $\Gamma_x$, for infrequent letters, in section A.1. (66) is obtained by following a method similar to the upper bound on $\Gamma_x$, for frequent letters, in section A.2.2. $\qquad\square$

We first the bound the sum of $T^n_{xy}$ over $(x,y)$ in $B(\delta)$ using (65).

$$
\begin{aligned}
\sum_{(x,y)\in B(\delta)} T^n_{xy} &\stackrel{(a)}{\leq} \sum_{(x,y)\in B(\delta)} 2\, q'_{xy}\, \frac{(\beta+2)}{\beta^5(n-1)} + \sum_{(x,y)\in B(\delta)} q_{xy}\,(1/n\beta^4)\left[1 + \frac{6}{\beta(n-1)}\right] \\
&\stackrel{(b)}{\leq} \Big(\sum_{(x,y)\in\mathcal{X}^2} q'_{xy}\Big)\left[2\,\frac{(\beta+2)}{\beta^5(n-1)}\right] + \Big(\sum_{(x,y)\in\mathcal{X}^2} q_{xy}\Big)\left[\frac{1}{n\beta^4} + \frac{6}{\beta^5 n(n-1)}\right] \\
&\stackrel{(c)}{\leq} O(1/n\beta^5),
\end{aligned}
\tag{67}
$$

where we get $(a)$ by using $\max_{p\in[0,1]} p(1-cp)^{n-2} < \frac{1}{c(n-1)}$, $(b)$ by using $\sum_{(x,y)\in B(\delta)} q_{xy} \leq \sum_{(x,y)\in\mathcal{X}^2} q_{xy}$, $\sum_{(x,y)\in B(\delta)} q'_{xy} \leq \sum_{(x,y)\in\mathcal{X}^2} q'_{xy}$, and $(c)$ because $\sum_{(x,y)\in\mathcal{X}^2} q_{xy}, \sum_{(x,y)\in\mathcal{X}^2} q'_{xy}$ are constant w.r.t. $n$.

Similarly, we get

$$
\sum_{(x,y)\in\overline{B}(\delta)} T^n_{xy} \leq O(n^3)\,\exp\left\{-(n-5)\,\frac{\beta^2}{2(\beta+2)}\,\delta\right\}
$$

using (66). Choosing $\delta = 8\,\frac{(\beta+2)}{\beta^2(n-5)}\ln n$, we have

$$
\sum_{(x,y)\in\overline{B}(\delta)} T^n_{xy} \leq O(1/n).
\tag{68}
$$

Combining (67) with (68), we get

$$
\sum_{x\in\mathcal{X}}\sum_{y\in\mathcal{X},y\neq x} T^n_{xy} \leq O(1/n\beta^5)
\tag{69}
$$

Combining (54), (58), (64), and (69) to upper bound (51) completes the proof of the upper bound on $R^*_n(P_{2,\beta})$ in Theorem 4.

## C   Proof of Theorem 4, Lower bound

To prove the lower bound on $R^*_n(\mathcal{P}_{2,\beta})$ in (10), we use the Le Cam method. The standard Le Cam method Yu (1997) is for estimating constant parameters of a distribution whereas $M_0(\boldsymbol{\pi}, X^n)$ is a function of both the distribution and the samples. To get (10), we use the following extension of Le Cam method for estimands that depend on both the distribution and its samples.

*Le Cam lower bound for estimating random variables*: Let $\mathcal{Q}$ be a family of distributions over an alphabet $\mathcal{Y}$ and $Y$ be a random variable distributed according to $Q \in \mathcal{Q}$. Let $\theta(Y, Q)$, taking values in a pseudometric space $\mathcal{D}$ with a pseudometric $d$, be a function of both $Y$ and the distribution $Q$. We assume that the set $\mathcal{D}$ is bounded i.e. the distance $d(u, v)$ between any two points $u, v \in \mathcal{D}$ is at most $\Delta$. Let $d(\mathcal{D}_1, \mathcal{D}_2) \triangleq \min_{u\in\mathcal{D}_1, v\in\mathcal{D}_2} d(u, v)$ be the distance between the subsets $\mathcal{D}_1, \mathcal{D}_2$ of $\mathcal{D}$. Let $\widehat{\theta}(Y)$ be an estimator for $\theta(Y, Q)$ and $co(\mathcal{Q})$ denote the convex hull of $\mathcal{Q}$.

The following lemma provides a lower bound on the worst-case risk (over $\mathcal{Q}$) of any estimator $\widehat{\theta}(Y)$ for $\theta(Y, Q)$.

**Lemma 12.** *(Chandra et al., 2022, Lemma 5) Let $\mathcal{D}_1, \mathcal{D}_2$ be two subsets of $\mathcal{D}$, and $\mathcal{Q}_1, \mathcal{Q}_2$ be two subsets of $\mathcal{Q}$ such that for any $Q_i \in \mathcal{Q}_i$, $\theta(Y, Q_i) \in \mathcal{D}_i$ with probability at least $1 - \epsilon_i$, $i = 1, 2$. Let $\delta \triangleq d(\mathcal{D}_1, \mathcal{D}_2)/2$, and $||Q_1 \wedge Q_2|| \triangleq 1 - ||Q_1 - Q_2||_{TV}$ denote the affinity of the two distributions $Q_1$ and $Q_2$. Then*

$$\sup_{Q \in \mathcal{Q}} E[d(\widehat{\theta}(Y), \theta(Y, Q))] \geq \delta \left( \sup_{Q_i \in co(\mathcal{Q}_i)} ||Q_1 \wedge Q_2|| \right) - (\epsilon_1 + \epsilon_2)\Delta. \tag{70}$$

To prove the lower bound (10) in theorem 4, we apply Lemma 12 on $\mathcal{P}_{2,\beta}$, the family of rank-2 diagonalizable t.p.ms with spectral gap $\beta$, using two specifically constructed t.p.ms $\in \mathcal{P}_{2,\beta}$ that are hard to distinguish.

We first prove (10) for $\beta \in (0, 1.6)$.

Let $P[\delta_1, \delta_2] \triangleq \begin{bmatrix} 1 - \delta_1 & (\delta_1/R)\, \mathbf{1}_{1 \times K-1} \\ \delta_2\, \mathbf{1}_{K-1 \times 1} & (1 - \delta_2)/R\, \mathbf{1}_{K-1 \times K-1} \end{bmatrix}$ be a $K \times K$ t.p.m parameterized by $\delta_1, \delta_2$, with $\mathbf{1}_{1 \times K-1}$ and $\mathbf{1}_{K-1 \times 1}$ denoting the row and column vectors with all entries as 1, and $\mathbf{1}_{K-1 \times K-1}$ denoting the $K - 1 \times K - 1$ matrix with all entries as 1.

Let $\mathcal{P}_1 = \{P_1\}$ and $\mathcal{P}_2 = \{P_2\}$ be two subsets of $\mathcal{P}_{2,\beta}$, where $P_1 = P[0.5\beta, 0.5\beta]$, $P_2 = P[0.5\beta - \alpha, 0.5\beta + \alpha]$ are two t.p.ms on the alphabet $\{1, 2, \ldots, L + 1\}$ with $\boldsymbol{\pi}_1 = 0.5 \begin{bmatrix} 1 & (1/L)\mathbf{1}_{1 \times L} \end{bmatrix}$ and $\boldsymbol{\pi}_2 = \begin{bmatrix} 0.5 + (\alpha/\beta) & (0.5 - (\alpha/\beta))/L\, \mathbf{1}_{1 \times L} \end{bmatrix}$ as their respective stationary distributions. For each element of $P_2$ to lie in $[0, 1]$ and each row of $P_2$ to sum up to 1, we require that $\alpha \leq \min\{0.5\beta, 1 - 0.5\beta\}$. Both $P_1$ and $P_2$ have the same spectral gap $\beta$ (and hence $\in \mathcal{P}_{2,\beta}$) and $\alpha, L$ are choosen appropriately to get the best lower bound on $R_n^*(\mathcal{P}_{2,\beta})$.

For $i = 1, 2$, if $X^n$ is a stationary Markov chain with t.p.m $P_i$, then $M_0(X^n, \boldsymbol{\pi}_i)$ satisfies

$$1 - M_0(X^n, \boldsymbol{\pi}_i) \overset{(a)}{\geq} \left(0.5 + (i-1)\frac{\alpha}{\beta}\right)\, I(N_1 \neq 0) \tag{71}$$

$$1 - M_0(X^n, \boldsymbol{\pi}_i) \overset{(b)}{\leq} \left(0.5 + (i-1)\frac{\alpha}{\beta}\right)\, I(N_1 \neq 0) + \frac{n}{L}\left(0.5 - (i-1)\frac{\alpha}{\beta}\right) \tag{72}$$

$$\text{i.e. } M_0(X^n, \boldsymbol{\pi}_i) \in \left[\left(0.5 - (i-1)\frac{\alpha}{\beta}\right)\left(1 - \frac{n}{L}\right), \left(0.5 - (i-1)\frac{\alpha}{\beta}\right)\right],$$

$$\text{with probability } 1 - \left(0.5 - (i-1)\frac{\alpha}{\beta}\right)\left(1 - \frac{\beta}{2} - (i-1)\alpha\right)^{n-1}, \tag{73}$$

where we get the bounds in $(a)$ and $(b)$ by considering the cases of : (i) the letter 1 occuring in all the samples $X^n$ and (ii) all the $n$ samples, $X^n$, being distinct with the letter 1 occurring only once, respectively. We get the confidence interval in (73) by using the bounds in (71) and (72) in the event of the letter 1 occurring atleast once.

Let $\alpha > 0.5\beta n/L$, so that the above confidence intervals in (73) for $M_0(X^n, \boldsymbol{\pi}_i), i = 1, 2$, are non-overlapping. Using lemma 12 with $\mathcal{Q} = \mathcal{P}_{2,\beta}, Y = X^n, \theta(Y, Q) = M_0(\boldsymbol{\pi}, X^n), \mathcal{D} = [0, 1], d(u, v) = (u - v)^2, \Delta = 1, \mathcal{Q}_i = \mathcal{P}_i$, and $D_i$ as the confidence interval in (73) for $i = 1, 2$, we get

$$\sup_{P \in \mathcal{P}} E[(\widehat{M_0}(X^n) - M_0(\boldsymbol{\pi}, X^n))^2] \geq 0.5 \left(\frac{\alpha}{\beta} - \frac{n}{2L}\right)^2 ||P_1(X^n) \wedge P_2(X^n)|| - \left(1 - \frac{\beta}{2}\right)^{n-1} \tag{74}$$

Our next lemma gives an upper bound on the total variation distance between $P_1(X^n)$ and $P_2(X^n)$.

**Lemma 13.** *For the t.p.ms $P_1, P_2 \in \mathcal{P}_{2,\beta}$, constructed above and $X^n$ being a stationary Markov chain,*

$$||P_1(X^n) - P_2(X^n)||_{TV} \leq (\sqrt{2}\alpha/\beta) \left(\frac{1 + 0.5(n-2)\beta}{1 - 0.5\beta}\right)^{0.5} \tag{75}$$

*Proof.* Section C.1. $\qquad\qquad\qquad\square$

Since $||P_1(X^n) \wedge P_2(X^n)|| = 1 - ||P_1(X^n) - P_2(X^n)||_{TV}$, using the above Lemma to lower bound the R.H.S in (74), we get

$$\sup_{P \in \mathcal{P}} E[(\widehat{M_0}(X^n) - M_0(\boldsymbol{\pi}, X^n))^2] \geq 0.5 \left(\frac{\alpha}{\beta} - \frac{n}{2L}\right)^2 \left(1 - (\sqrt{2}\alpha/\beta) \left(\frac{1 + 0.5(n-2)\beta}{1 - 0.5\beta}\right)^{0.5}\right)$$
$$- \left(1 - \frac{\beta}{2}\right)^{n-1} \tag{76}$$

Substituting $\alpha = \frac{\beta}{2\sqrt{2}} \left(\frac{1 - 0.5\beta}{1 + 0.5(n-2)\beta}\right)^{0.5}$, and $L = e^n$ in (76) completes the proof of (10) for $\beta \in (0, 1.6)$.

We now prove (10) for $\beta \in [1.6, 2]$. Let $\mathcal{P}'_1 = \{P'_1\}$ and $\mathcal{P}'_2 = \{P'_2\}$ be two subsets of $\mathcal{P}_{2,\beta}$, with $\beta \in [1.6, 2]$, where
$$P'_1 = \begin{bmatrix} 2 - \beta & 0.5(\beta - 1) & (0.5(\beta - 1)/L) \, \mathbf{1}_{1 \times L} \\ \mathbf{1}_{1 \times L+1} & \mathbf{0}_{L+1 \times L+1} \end{bmatrix},$$
$$P'_2 = \begin{bmatrix} 2 - \beta & 0.5(\beta - 1) + \alpha' & ((0.5(\beta - 1) - \alpha')/L) \, \mathbf{1}_{1 \times L} \\ \mathbf{1}_{1 \times L+1} & \mathbf{0}_{L+1 \times L+1} \end{bmatrix}$$ are two t.p.ms on the alphabet
$\{1, 2, \ldots, L + 2\}$ with $\boldsymbol{\pi}'_1 = \begin{bmatrix} 1/\beta & 0.5(\beta - 1)/\beta & 0.5(\beta - 1)/\beta L \, \mathbf{1}_{1 \times L} \end{bmatrix}$ and
$\boldsymbol{\pi}'_2 = \begin{bmatrix} 1/\beta & (0.5(\beta - 1) + \alpha')/\beta & (0.5(\beta - 1) - \alpha')/\beta L \, \mathbf{1}_{1 \times L} \end{bmatrix}$ as their respective stationary distributions and $\mathbf{0}_{L+1 \times L+1}$ being the $L + 1 \times L + 1$ all zero matrix. For each element of $P'_2$ to lie in $[0, 1]$ and each row of $P'_2$ to sum up to 1, we require that $\alpha' \leq \min\{0.5(\beta - 1), 1 - 0.5(\beta - 1)\}$. Both $P'_1$ and $P'_2$ have the same spectral gap $\beta$ (and hence $\in \mathcal{P}$) and $\alpha', L$ are choosen appropriately to get the best lower bound on $R^*_n(\mathcal{P}_{2,\beta})$.

Following a method similar to the proof of (10) for $\beta \in (0, 1.6)$, we get

$$\sup_{P' \in \mathcal{P}'} E[(\widehat{M_0}(X^n) - M_0(\boldsymbol{\pi}, X^n))^2] \geq \frac{1}{2} \left(\frac{\alpha'}{\beta} - \frac{n}{2L\beta}(\beta - 1)\right)^2 ||P'_1(X^n) \wedge P'_2(X^n)|| - 9\,(0.5)^{\lfloor \frac{n-3}{2} \rfloor} \tag{77}$$

Our next lemma gives an upper bound on the total variation distance between $P'_1(X^n)$ and $P'_2(X^n)$.

**Lemma 14.** *For the t.p.ms $P'_1, P'_2 \in \mathcal{P}_{2,\beta}$, constructed above and $X^n$ being a stationary Markov chain,*

$$||P'_1(X^n) - P'_2(X^n)||_{TV} \leq \sqrt{2n}\alpha'/\sqrt{\beta(\beta - 1)} \tag{78}$$

*Proof.* Section C.1. $\qquad\square$

Using the above lemma to lower bound the R.H.S in (77) and choosing $\alpha' = 0.5\sqrt{\beta(\beta - 1)}/\sqrt{2n}$, $L = e^n$ completes the proof of (10) for $\beta \in [1.6, 2]$.
This completes the proof of the lower bound on $R^*_n(\mathcal{P}_{2,\beta})$.

## C.1 Proof of Lemmas 13, 14

To show the bound in (75), we first bound the total variation distance between $P_1(X^n)$ and $P_2(X^n)$ by the KL divergence $D_{KL}(P_1(X^n)||P_2(X^n))$ using Pinsker's inequality.

**Lemma 15.** *Pinkser's inequality (Boucheron et al., 2013, Theorem 4.19)*

$$||P_1(X^n) - P_2(X^n)||_{TV} \leq \frac{1}{\sqrt{2}} \sqrt{D_{KL}(P_1(X^n)||P_2(X^n))} \tag{79}$$

*where $D_{KL}(P_1(X^n)||P_2(X^n))$ is the KL divergence between $P_1(X^n)$ and $P_2(X^n)$.*

Our next lemma expresses $D_{KL}(P_1(X^n)||P_2(X^n))$ in terms of the KL divergence between the stationary distributions $\boldsymbol{\pi}_1, \boldsymbol{\pi}_2$ and the KL divergence between the corresponding rows of $P_1$ and $P_2$.

**Lemma 16.** *For any two t.p.ms $P_1, P_2$, on an $\mathcal{X}$, with $\boldsymbol{\pi}_1, \boldsymbol{\pi}_2$ as their respective stationary distributions,*

$$D_{KL}(P_1(X^n)||P_2(X^n)) \;=\; D_{KL}(\boldsymbol{\pi}_1||\boldsymbol{\pi}_2) + (n-1) \sum_{x \in \mathcal{X}} \pi_{1,x} \, D_{KL}(P_1(\cdot|x) \,||\, P_2(\cdot|x)) \tag{80}$$

*where $P_i(\cdot|x)$ denotes the row of the t.p.m $P_i, i = 1, 2$, with transition probabilities from the state $x$.*

*Proof.* Let $x^n \triangleq (x_1, x_2, \ldots, x_n) \in \mathcal{X}^n$.

$$
\begin{aligned}
&D_{KL}(P_1(X^n)||P_2(X^n)) \\
&= \sum_{x^n \in \mathcal{X}^n} P_1(x^n) \, \ln(P_1(x^n)/P_2(x^n)) \\
&\overset{(a)}{=} \sum_{x^n \in \mathcal{X}^n} P_1(x^n) \left[ \ln\left(\pi_{1,x_1}/\pi_{2,x_1}\right) + \sum_{l=2}^{n} \ln \frac{P_1(X_l = x_l | X_{l-1} = x_{l-1})}{P_2(X_l = x_l | X_{l-1} = x_{l-1})} \right] \\
&\overset{(b)}{=} D_{KL}(\boldsymbol{\pi}_1||\boldsymbol{\pi}_2) + \sum_{l=2}^{n} \sum_{x^l \in \mathcal{X}^l} P_1(x^l) \, \ln \frac{P_1(X_l = x_l | X_{l-1} = x_{l-1})}{P_2(X_l = x_l | X_{l-1} = x_{l-1})} \\
&\overset{(c)}{=} D_{KL}(\boldsymbol{\pi}_1||\boldsymbol{\pi}_2) \\
&\quad + \sum_{l=2}^{n} \sum_{x_{l-1}, x_l \in \mathcal{X}} \pi_{1,x_{l-1}} P_1(X_l = x_l | X_{l-1} = x_{l-1}) \ln \frac{P_1(X_l = x_l | X_{l-1} = x_{l-1})}{P_2(X_l = x_l | X_{l-1} = x_{l-1})} \\
&= D_{KL}(\boldsymbol{\pi}_1||\boldsymbol{\pi}_2) + (n-1) \sum_{x \in \mathcal{X}} \pi_{1,x} \, D_{KL}(P_1(\cdot|x) \,||\, P_2(\cdot|x))
\end{aligned}
$$

where we get $(a)$ by using the Markov property $P_i(x^n) = \pi_{i,x_1} \prod_{l-2}^{n} P_i(X_l = x_l | X_{l-1} = x_{l-1})$, $i = 1, 2$, $(b)$ and $(c)$ by appropriately marginalizing $P_1(x^n)$. $\qquad\square$

### C.1.1 Proof of Lemma 13

Using the values specified for $\pi_{i,x}$, $P_i(X_2 = y | X_1 = x)$ for $x, y \in \{1, \ldots, L+1\}$, $i = 1, 2$, in the section C, we get

$$
\begin{aligned}
D_{KL}(\boldsymbol{\pi}_1||\boldsymbol{\pi}_2) &= -0.5 \ln\left(1 - 4\alpha^2/\beta^2\right) \\
D_{KL}(P_1(\cdot|1) \,||\, P_2(\cdot|1)) &= -(1 - 0.5\beta) \, \ln\left(1 + \alpha/(1 - 0.5\beta)\right) - 0.5 \, \beta \, \ln\left(1 - 2\alpha/\beta\right) \\
D_{KL}(P_1(\cdot|x) \,||\, P_2(\cdot|x)) &= -(1 - 0.5\beta) \, \ln\left(1 - \alpha/(1 - 0.5\beta)\right) - 0.5 \, \beta \, \ln\left(1 + 2\alpha/\beta\right), \\
&\quad \text{for } x \in \{2, \ldots, L+1\}.
\end{aligned}
$$

Using the above three equations in (80), we get

$$
\begin{aligned}
&D_{KL}(P_1(X^n)||P_2(X^n)) \\
&= -0.5 \, (n-1) \left[ (1 - 0.5\beta) \, \ln\left(1 - \alpha^2/(1 - 0.5\beta)^2\right) + 0.5 \, \beta \, \ln\left(1 - 4\alpha^2/\beta^2\right) \right] \\
&\quad - 0.5 \ln\left(1 - 4\alpha^2/\beta^2\right) \\
&\overset{(a)}{\leq} (n-1) \left[ \frac{\alpha^2}{1 - 0.5\beta} + 2\frac{\alpha^2}{\beta} \right] + 4\frac{\alpha^2}{\beta^2} \;=\; 4\frac{\alpha^2}{\beta^2(1 - 0.5\beta)} \, [1 + 0.5(n-2)\beta]
\end{aligned}
$$

where we get $(a)$ by using $-\ln(1 - x) \leq 2x$, for $x \in (0, 0.5)$. Plugging the above bound into (79) completes the proof of Lemma 13.

### C.1.2   Proof of Lemma 14

Using the values specified for $\pi'_{i,x}$, $P'_i(X_2 = y | X_1 = x)$ for $x, y \in \{1, \ldots, L+2\}$, $i = 1, 2$, in the section C, we get

$$
D_{KL}(\boldsymbol{\pi}_1 || \boldsymbol{\pi}_2) = -\frac{(\beta - 1)}{2\beta} \ln\left[1 - \left(\frac{2\alpha'}{\beta - 1}\right)^2\right]
$$

$$
D_{KL}(P'_1(\cdot|1) || P'_2(\cdot|1)) = -0.5(\beta - 1) \ln\left[1 - \left(\frac{2\alpha'}{\beta - 1}\right)^2\right]
$$

$$
D_{KL}(P'_1(\cdot|x) || P'_2(\cdot|x)) = 0, \quad \text{for } x \in \{2, \ldots, L+2\}.
$$

Using the above three equations in (80) and following a method similar to C.1.1, we get $||P'_1(X^n) - P'_2(X^n)||_{TV} \leq \sqrt{2n}\alpha'/\sqrt{\beta(\beta - 1)}$. This completes the proof of Lemma 14.

