# OpenReview forum: "How good is Good-Turing for Markov samples?"
_TMLR — Accepted by TMLR_

### Review · Reviewer_25AQ · 2023-12-18

**Summary Of Contributions:**

This paper considers a classical problem of missing mass estimation via the Good--Turing estimator but for Markov samples, instead of the usual iid case. Bounds on the bias and squared error risk are derived and shown to be dependent on the convergence rate of $(P^{\sim x})^n$ over all $x$ that contribute to the missing mass. Additionally, for rank-2 Markov chains, the authors show bounds on the minimax risk that depend on the spectral gap $\beta$.

**Audience:**

Yes

**Broader Impact Concerns:**

NIL

**Claims And Evidence:**

Yes

**Requested Changes:**

Please consider the comments above.

**Strengths And Weaknesses:**

Strengths:

- I believe that the work certainly contains novel contributions. To the best of my knowledge, missing mass estimation has primarily been done for iid samples. The extension to Markov samples is of practical interest as most data cannot be modelled as an iid stochastic process, so the Markovian assumption brings us closer to reality.

- The paper is also fairly well written and the results are easy to understand. It is clear that the authors have put in a great deal of effort to explain their contributions as lucidly as possible.

- The bounds, while not tight for the case of rank-2 Markov chains, are the first step to understanding this problem and serve as a starting point for others to build on. The proof techniques are rather novel, from what I can tell.

Weaknesses:

- The reviewer did not really get how "a chain with non-zero, constant spectral gap" results in the GT estimator failing to converge. Presumably, this is done via the experiments, but using the experiments to illustrate this counterexample is not particularly illuminating. Can the authors use an analytical example to make this point?

- The authors also claim that they demonstrate through examples that the GT estimator fails to converge to the true missing mass for Markov chains "whose mixing times are 2 or 3". First, the notion of "mixing time" has not been formally defined. There are different definitions for this, and they depend on some $\epsilon$, Second, what is so special about the (integer) numbers 2 and 3? Mixing times can be non-integer-valued. Third, the demonstrations are again empirical. Having an analytical demonstration or counterexample would be clearer.

- Of course, there is a big gap between the minimax upper and lower bounds; they depend on $1/\beta^5$ and $1/\beta$ respectively. I suppose the authors conjecture that the upper bound is loose due to their experiment in Section 3.3.1. How can it be strengthened?

- I would usually imagine that the chain is fixed so that $\beta$ is constant. It is rather strange that the authors discuss the case that $\beta$ varies with $n$ below Theorem 3. What one would do to find the right dependence on $\beta$ is to consider several chains, with different $\beta$. Run the GT estimator on each of the chains. Estimate the dependence of the minimax risk on $\beta$ from the different experiments. Currently, the authors are looking at a single chain parametrized by $\beta$ and varying $\beta$ as a function of $n.

---

### Review · Reviewer_Yr9V · 2024-01-15

**Summary Of Contributions:**

Consider the problem of estimating the missing mass of a Markov chain given random variables $X_1, \ldots, X_n$ following the chain. This paper derives estimation error bounds for the Good-Turing (GT) estimator, showing that they recover the known $O ( 1 / n )$ rate for the i.i.d. case and providing sufficient conditions for the error bounds to vanish as $n \to \infty$. In addition, for the special case where the transition matrix is of rank 2, this paper provides a $\Omega ( 1 / (n \beta) )$ error lower bound and proves that the GT estimator satisfies a $O ( 1 / ( n \beta^5 ) )$ error upper bound, where $\beta$ denotes the spectral gap.

**Audience:**

Yes

**Broader Impact Concerns:**

N/A. This is a theory work.

**Claims And Evidence:**

No

**Requested Changes:**

- Please justify studying the standard GT estimator for the Markov case.
- Please justify the definition of the missing mass studied in this paper and its significance.
- Please elaborate on the meaning of Lemma 1.
- p. 2: Please provide more information about the "forays towards missing mass estimation from Markov chains." In particular, please compare them with this paper.
- Theorem 2:
    - Please provide the proof details and check if there are missing assumptions on $n$.
    - Please justify the conditions on the transition matrix $P$.
- Typos:
    - p. 2: worst case -> worst-case.
    - p. 3: how GT estimator... -> how the GT estimator...

**Strengths And Weaknesses:**

Strengths.
- The problem studied is fundamental.
- The paper is quite complete in the sense that it provides estimation error bounds, characterizes the minimax error for rank-2 Markov chains, and conducts empirical studies on real data.

Weaknesses.
- I wonder if the problem formulation is appropriate.
    - Considering that most known results for the Good-Turing estimator pertain to the i.i.d. case, it also appears natural, if not more natural, to propose a generalized estimator in the Markov case that reduces to the GT estimator in the i.i.d. case.
    - I am not an expert of this topic. It appears that there are other definitions of the missing mass for the Markov case. For instance, the definition of missing mass proposed by Skorski (2020) also looks reasonable
    - This paper justifies the significance of missing mass estimation in Markov chains only via citing existing papers for the i.i.d. case. Please verify that those significances carry over to the Markov case.
- The correctness of Theorem 2 is unclear. After establishing the bound $E[ \hat{M}_0^{\text{GT}} ( X^n ) ] - E [ M_0 ( \pi, X^n ) ] \leq c$ for some $c > 0$, the paper claims that the bound $E [ M_0 ( \pi, X^n ) ] - E[ \hat{M}_0^{\text{GT}} ( X^n ) ] \leq c$ can be similarly proved, yet provides no details. The derivation of the second inequality is not apparent.
- The meaning and necessity of Lemma 1 are unclear to me. Lemma 1 is not used for deriving later lemmas and theorems. Lemma 1 surely justifies the GT estimator for the i.i.d. case, but it is already known that the GT estimator is good for the i.i.d. case.
- Assumptions in Theorem 2: Regarding Theorem 2, it is unclear if the condition $n > n_0$ is required, as well as $n > 4$ for the first bound and $n > 6$ for the second bound.
- Theorem 2: It appears unclear why "we expect such left and right eigenvectors of $P^{\sim x}$ to be close to those of $1 \pi^{\sim x}$ and the Perron eigenvalue of $P^{\sim x}$ to be close to $1 - \pi_x$".
- Comparison with exising literature does not appear to be complete. (See the requested change for p.2 below.)

---

### Review · Reviewer_3aVP · 2024-01-22

**Summary Of Contributions:**

This paper addresses the problem of missing mass estimation in a Markovian setting. Namely, from a single trajectory of length $n$ sampled from an unknown transition matrix $P$, the goal is to estimate the stationary mass of yet unobserved symbols.

The authors consider the popular Good-Turing (GT) estimator, which has been analyzed extensively in the iid setting.

Contributions:
1. Propose sufficient conditions for the GT estimator to converge to the missing stationary mass and bound the absolute bias of the GT estimator in terms of some properties of the transition matrix.
2. Propose an analysis of the minimax sample complexity over the class of rank 2 Markov chains with prescribed spectral gap.
3. Propose experiments based on both synthetic data and existing text corpora.

**Audience:**

Yes

**Claims And Evidence:**

No

**Requested Changes:**

**Major**

1. Increase the discussion on prior work on the topic of missing mass estimation in the Markovian setting. What estimators were considered? What was their limitation? How does the present study improve upon them? For instance, how do the approaches, assumptions and results compare with Chandra et al. [2020, 2022], Skorski [2020]?

2. Are there connections between the missing mass problem and the notion of surprise probabilities analyzed in Norris et al. [2014]? If so, could the authors discuss these connections?

3. The idea that the rank of the transition matrix should control the minimax sample complexity is not convincing. The rank is a brittle property. A randomly infinitesimally perturbed uniform matrix (iid) will have almost surely full-rank while we should not expect the statistical problem to become significantly larger. This makes the definition of the minimax problem over this class of chain rather unnatural.

4. In Theorem 4: can the authors make precise the statement “for $n$ sufficiently large”? Is the constant $c$ universal?

5. Can the authors put forward natural reasons for a transition matrix to enjoy rank 2 (beyond mathematical convenience)? Is there a probabilistic interpretation? For instance, reversibility of a Markov chain is also brittle, but many physical processes obeying are thought to be reversible in nature.

6. At the top of page 3, the authors argue that taking a thinned Markov trajectory, where observations are spaced by the order of the mixing time, would overestimate the missing mass. Leaving out the problem of obtaining a bound on the mixing time, sub-sampling is not a necessity for estimation problems; the mixing time or the inverse of the spectral gap often simply enters the sample complexity multiplicatively. See for instance Paulin [2015, Proposition 3.16.] for the problem of estimating the stationary distribution.

**Minor**

1. On page 3, the authors seems to make a distinction between behaviors they would expect for small mixing times and large spectral gaps. However, at least in the reversible setting, both are known to be closely related. It is not clear why the authors have decided to make such a distinction.

2. On page 3, could the authors give a reference as to how to obtain $\sum_{x \in \mathcal{X}} \pi_x^2(1 - \pi_x)^{n-1} = O(1/n)$. For
instance, please point to some analysis in the iid setting.

3. The minimax problem in framed in terms of a stationary start. Could the authors comment on this assumption, and how it might be relaxed to an arbitrary starting state?

4. The authors should consider moving Section 5, Section 6 and Section 7 to the appendix and dedicate a section to summarizing the SOTA in the Markovian setting.

5. Please only number equations which are referred to somewhere in the manuscript.

6. Small typo: p.8 in the footnote “Pubilc”.

**References**

P. Chandra, A. Thangaraj, and N. Rajaraman. Missing mass of markov chains. In 2020 IEEE International Symposium on Information Theory (ISIT), pages 1207–1212. IEEE, 2020.

P. Chandra, A. Thangaraj, and N. Rajaraman. Missing mass estimation from sticky channels. In 2022 IEEE International Symposium on Information Theory (ISIT), pages 910–915. IEEE, 2022.

J. Norris, Y. Peres, and A. Zhai. Surprise probabilities in markov chains. In Proceedings of the Twenty-Sixth Annual ACM-SIAM Symposium on Discrete Algorithms, pages 1759–1773. SIAM, 2014.

D. Paulin. Concentration inequalities for markov chains by marton couplings and spectral methods. 2015.

M. Skorski. Missing mass concentration for markov chains. arXiv preprint arXiv:2001.03603, 2020.

**Strengths And Weaknesses:**

**Strengths**
1. Contribute towards a fundamental problem in statistics.
2. Illustration of theoretical claims with numerical evidence.

**Weaknesses**
1. The discussion on the relevant literature is largely insufficient.
2. Conditions in Theorem 2 seem rather artificial and are challenging to verify in general. This is acknowledged by the authors at the beginning of Section 3.3.3.
3. The argument proposing that the rank of the transition matrix captures the minimax sample complexity is not convincing.

Please refer to the comments in the “Requested changes” section for more details.

---

### Decision · Action_Editor_kfep · 2024-03-24

**Recommendation:** Accept with minor revision

**Comment:**

There is one minor concern raised by Reviewer 3aVP, that the authors should address in their revision. I am copying the reviewer's comments below:

> The authors make the following claim: "The minimax rate of missing mass estimation for Markov chains will depend on the rank of the t.p.m." This claim is backed up by the experiment which is illustrated in Figure 2 (right), for the GT estimator. In their experiment, the authors construct a family of chains whose rank can be controlled by a parameter L. They show that increasing L leads to different decays of the MSE with respect to the sample size. However, besides increasing the rank in their construction, they are also changing other properties of the chain, so that it is difficult to see that it is really the rank that matters.

Please acknowledge and clarify this point in your revision.

**Audience:**

Yes, to quote one reviewer, "The problem under consideration is undoubtedly interesting and should be of interest to the community of TMLR".

**Claims And Evidence:**

This paper considers the problem of estimating the missing mass of a Markov chain: Given a single trajectory sampled from a Markov chain with unknown transition matrix $P$, the goal is to estimate the stationary mass of the unobserved symbols. The authors derive error bounds for the Good-Turing (GT) estimator, showing that it recovers the known 1/n rate for the i.i.d. case and for the special case where the transition matrix is of low rank, they provide upper and lower bounds that nearly match (i.e. the gap is in terms of the spectral gap, but optimal in the sample size.